# BRIDGING FORMAL LANGUAGE WITH CHAIN-OF-THOUGHT REASONING TO GEOMETRY PROBLEM SOLVING

## ABSTRACT

Large vision language models exhibit notable limitations on Geometry Problem Solving (GPS) because of their unreliable diagram interpretation and pure natural-language reasoning. A recent line of work mitigates this by using symbolic solvers: the model directly generates a formal program that a geometry solver can execute. However, this direct program generation lacks intermediate reasoning, making the decision process opaque and prone to errors. In this work, we explore a new approach that integrates Chain-of-Thought (CoT) with formal language. The model interleaves natural language reasoning with incremental emission of solver-executable code, producing a hybrid reasoning trace in which critical derivations are expressed in formal language. To teach this behavior at scale, we combine (1) supervised fine-tuning on an 11K newly developed synthetic dataset with automatic formalization and interleaved formal-natural language reasoning, and (2) solver-in-the-loop reinforcement learning that jointly optimizes both the CoT narrative and the resulting program through outcome-based rewards. Built on Qwen2.5-VL-7B, our new model, named GF-Reasoner, achieves up to 15% accuracy improvements on standard GPS benchmarks, surpassing both 7B-scale peers and the much larger model Qwen2.5-VL-72B. By exploiting high-order geometric knowledge and offloading symbolic computation to the solver, the generated reasoning traces are noticeably shorter and cleaner. Furthermore, we present a comprehensive analysis of method design choices (e.g., reasoning paradigms, data synthesis strategies, training methodologies, etc.), providing actionable insights for future research.

## 1 INTRODUCTION

Large Vision Language Models (LVLMs) have emerged as powerful tools for a wide range of applications, demonstrating impressive capabilities in tasks such as visual question answering, video understanding, etc. (Li et al., 2023; Liu et al., 2023; Team et al., 2024; Hurst et al., 2024; Lu et al., 2024a; Bai et al., 2025). Despite these rapid advances, LVLMs still exhibit significant weaknesses in reasoning ability (Chen et al., 2021; Lu et al., 2024b; Zhang et al., 2024). When confronted with tasks requiring spatial or geometric reasoning, current models frequently produce inconsistent or incorrect results, substantially limiting their utility in practical applications (Liu et al., 2023; Gupta & Kembhavi, 2023; Yang et al., 2024).

This paper investigates Geometry Problem Solving (GPS) (Chen et al., 2023; Lu et al., 2024b), a particularly challenging reasoning task where LVLMs must reason across geometric diagrams while applying geometry knowledge. Current research to improve LVLM reasoning capabilities has focused predominantly on *natural language* reasoning approaches (Bai et al., 2025; Team et al., 2025; Huang et al., 2025), which, however, are limited in handling numerical computations, often produce redundant outputs, and cannot ensure the solution process is correct (see Figure 1(a)).

To tackle this, recent works have investigated the application of *formal language* (Lu et al., 2021; Zhang et al., 2023b;a; Xia et al., 2024). Given textual questions and accompanying diagrams, LVLMs are trained to predict formal programs encoding solver-executable solution steps. The programs are subsequently executed by a geometry solver to derive numerical solutions (see Figure 1(b)). However, the direct program generation approach lacks the flexibility in performing inter-

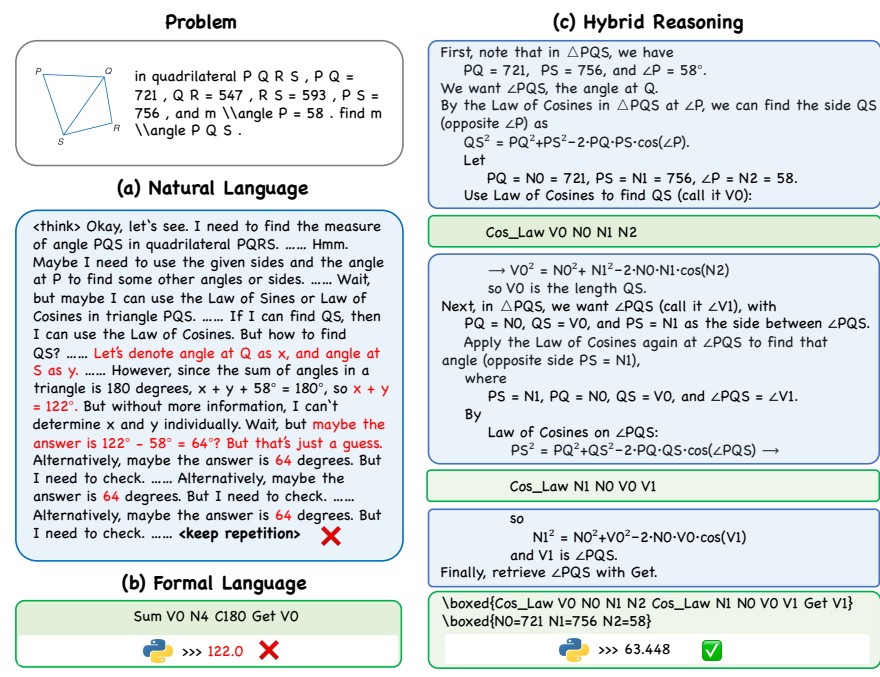

Figure 1: Illustration of interleaved formal-natural reasoning for geometry problem solving. (a) Natural language solution from Vision-R1, in which the symbol "..." denotes the omitted long reasoning steps; (b) Formal language solution generated by GeoX (Xia et al., 2024); (c) Hybrid reasoning solution produced by our GF-Reasoner model.

mediate steps, which fundamentally restricts both the interpretability of its decision-making process and the model's reasoning capability. This raises a key research question:

*How can we bridge intermediate reasoning with formal language in geometry problem solving?*

In this work, we propose a hybrid reasoning framework that synergizes the flexibility of natural language reasoning with the precision of formal language. As shown in Figure 1(c), our framework interleaves natural language reasoning with the progressive generation of formal programs. Within this framework, natural language handles diagram interpretation, problem formalization, and reasoning trajectory planning. Meanwhile, formal programs implement geometric theorems as executable operators with explicit variable bindings during critical steps. This integration equips LVLMs with both general and rigorous reasoning capabilities for geometry problems.

How can we implement the above framework in practice? We find that simple prompting fails to achieve the desired results (see Section 3.1), primarily because the formal language syntax and interleaved formal-natural reasoning pattern may not be sufficiently represented within the model's internal knowledge base. Therefore, we turn to post-training strategies to teach the model to perform hybrid reasoning. This approach presents several technical challenges. First, such training data is scarce online, raising critical questions: How can we construct an effective hybrid reasoning dataset, and what essential characteristics should it possess? Second, determining proper training strategies remains challenging. Our experience indicates that straightforward supervised fine-tuning fails to unlock hybrid reasoning potential and cannot match natural language reasoning performance.

To answer the above questions, we employ two scalable post-training strategies: 1) *SFT on a newly developed Interleved Formal-Natural Chain-of-Thought (IFN-CoT) dataset.* We curate an 11k-sample CoT dataset featuring auto-formalization and interleaved formal-natural language reasoning trajectories. The dataset is constructed via bidirectional synthesis to enhance the utilization efficiency of available data. SFT on this dataset enables the model to automatically formalize informal inputs into formats suitable for formal reasoning (e.g., by explicitly binding problem and process variables to operands) and internalize geometric formal language syntax. 2) *Solver-integrated RL.* SFT alone proves insufficient for achieving a strong hybrid reasoning capacity (see Section 4.3).

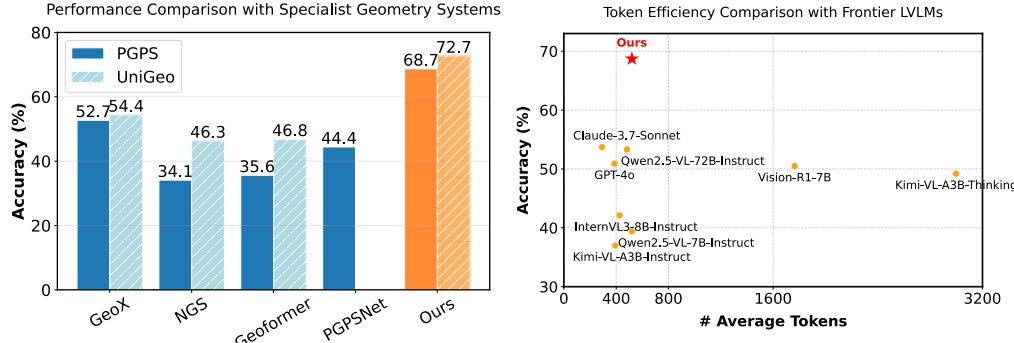

Figure 2: Left: Performance comparison with specialist geometry systems (GeoX, NGS, Geoformer, and PGPSNet) using direct formal program generation, evaluated on two benchmarks (PGSP and UniGeo). Right: Token efficiency comparison with frontier LVLMs.

Therefore, we develop a solver-in-the-loop RL framework where a geometric solver executes the resulting formal program and returns verification feedback on solution correctness. This iterative process refines the CoT trajectory and ultimately increases performance by a substantial margin.[1]

Built upon Qwen2.5-VL-7B-Instruct (Bai et al., 2025), our model, GF-Reasoner (Geometric Formal Reasoner), demonstrates superior performance over both specialized geometry systems (e.g., GeoX (Xia et al., 2024), PGPSNet (Zhang et al., 2023a)) and state-of-the-art multimodal LLMs (e.g., Claude-3.7 Sonnet, Qwen2.5-VL-72B (Bai et al., 2025)), as shown in Figure 2.[2] We can also notice that our model has the additional benefits of concise outputs, achieved by condensing complex solving processes into compact operators and offloading explicit computation to an external solver. In line with these results, we also present a comprehensive analysis of our framework designs spanning from the reasoning paradigm, data synthesis strategy, to training methodologies.

Our main contributions can be summarized as follows:

- We propose a new framework that integrates CoT with formal language for GPS, a new hybrid reasoning paradigm that combines the strengths of both approaches.
- We curate a new 11k-sample interleaved formal-natural CoT dataset using a bidirectional synthesis methodology. This is the first dataset of its kind, which enables us to explore post-training procedures such as SFT and RL to teach LVLMs external knowledge of geometric formal language.
- We empirically validate the effectiveness of integrating CoT with formal language, demonstrating improvements in both performance and token efficiency. We also provide comprehensive studies that may provide several actionable insights for future research.

## 2  RELATED WORK

**Geometry Problem Solving (GPS).** GPS plays a crucial role in industry, manufacturing, and scientific areas. Since the 1970s, automated GPS has been an important research focus, which includes geometry calculation and geometry proving. In the context of geometry calculation, existing approaches fall into two categories: symbolic solvers and neural solvers. Symbolic solvers, such as GEOS (Seo et al., 2015), Inter-GPS (Lu et al., 2021), and E-GPS (Wu et al., 2024), parse diagrams and textual problems into several conditions. They then iteratively deduce new conditions by applying theorem rules until the problem is solved. While these solvers offer high interpretability, they rely on meticulously designed rules, making them difficult to extend. Recent advances in AI have shifted the focus from symbolic to neural methods. By leveraging neural networks to interpret geometry diagrams and derive formal program solutions, models like NGS (Chen et al., 2021),

---

[1]Our empirical results show that RL increases Pass@1 accuracy by 26%. Notably, without RL, our model fails to match the performance of natural language reasoning alone.

[2]PGPSNet is not evaluated on UniGEO as the benchmark lacks annotations for structural and semantic clauses required by the model.

Table 1: Comparison with existing methods in geometry problem solving.

| Method | Language | CoT Reasoning |
|---|---|---|
| Specialist Geometry System | Formal Language | × |
| MLLMs | Natural Language | ✓ |
| GF-Reasoner (Ours) | Interleved Formal-Natural | ✓ |

Geoformer (Chen et al., 2022), PGPSNet (Zhang et al., 2023a), and GeoX (Xia et al., 2024) have demonstrated promising results. These approaches use neural networks to parse the problem and predict a formal-language solution, which is then executed by an external solver to compute the numerical answer.

Current neural methods have several notable limitations. On the one hand, they directly predict the result without intermediate reasoning, constraining their ability to solve more complex problems. On the other hand, some specialist geometry models face limited adaptability due to their restrictive input requirements. For example, GeoX, NGS and Geoformer require problem variables to be explicitly declared in text questions (e.g., the text questions are like "In triangle ABC, AC = N0, AB = N1, ...", where the value of N0 and N1 are predefined in a numerical list). In contrast to existing methods, our work is the first to integrate geometric formal language with CoT reasoning, possibly unlocking the reasoning potential of neural approaches for geometry calculation problems. Meanwhile, our method can auto-formalize the informal inputs by automatically binding problem variables into operands, offering higher flexibility.

**Multimodal Large Language Model Reasoning.** Improving complex reasoning capability of large models has been regarded as a critical pathway toward artificial general intelligence (Guo et al., 2025). In the field of large multimodal models, Zhang et al. (2024) are among the first to systematically evaluate the mathematical reasoning ability of LVLMs. Their results reveal that LVLMs often generate incorrect answers with an incorrect reasoning process. Furthermore, among incorrect reasoning, calculation errors could contribute to 19.95%. GeoSense (Xu et al., 2025a) is a recently developed benchmark to systematically evaluate the geometric reasoning abilities of LVLMs through the lens of geometric principles. They found that the identification and application of geometric principles remain a bottleneck for leading LVLMs. Recent advances in multimodal reasoning have led to several notable approaches (Huang et al., 2025; Xu et al., 2025b; Luo et al., 2025; Shen et al., 2025; Wang et al., 2025). A representative example is Vision-R1 (Huang et al., 2025), which employs DeepSeek R1 to automatically generate visual reasoning data from textual descriptions. While these methods demonstrate promising results, they primarily rely on natural language reasoning. In our work, we overcome natural language's imprecision and redundancy by combining it with formal language, enabling both flexible and concise geometric reasoning.

## 3 METHOD

This section presents our method for GPS. We begin by providing background on geometric formal language in Section 3.1, followed by our data synthesis approaches in Section 3.2, and finally present the training procedures in Section 3.3.

### 3.1 GEOMETRIC FORMAL LANGUAGE

Our work builds on the geometry solver and formal language defined in (Zhang et al., 2023a). It has 34 *operators* and 55 *operands*.[3] Each operator encodes a specific geometric theorem or axiom, covering fundamental operations across triangles, quadrilaterals, polygons, circles, and other geometric shapes. These operators mainly work with three types of operands: *problem variables* (N) representing known measurements from the problem statement, *process variables* (V) representing intermediate results generated during computation, and *constants* (C) encoding common numerical values. A formal program specifies a deduction step using these operators and operands. For example, the formal program `Gougu N0 N1 V0` applies the Pythagorean theorem to calculate the hypotenuse, where `N0` and `N1` are two known leg lengths, and `V0` is the hypotenuse to be computed.

---

[3]Compared to geometry solvers used in prior work (Chen et al., 2021; 2022), this formal language includes 16 additional operators and provides broader coverage of geometric theorems.

To leverage the geometry solver, the model must first understand the diagram and text question, define variables as operands (formalization), deduce relationships using geometry operators (reasoning), and finally construct the formal program (coding). After generation, both the program and extracted problem variables are sent to the geometry solver to compute numerical results for unknown process variables. This executor performs symbolic algebraic calculations to solve for unknown variables within the program's formulas, implemented via Python's SymPy library.

This framework offers two key advantages. First, it decouples reasoning from numerical computation by offloading arithmetic calculations to an external solver, eliminating calculation errors common in natural language reasoning involving complex operations (e.g., inverse trigonometric functions). Second, it provides significant conciseness over natural language—while natural language requires multi-step descriptions (e.g., calculate the octagon's area by dividing into triangles, computing each area, then summing), formal language expresses this as: `RNgon_B_Area C8 N0 V0` (where `N0` = side length, `V0` = resulting area). This conciseness helps mitigate reasoning errors that arise from the long contexts required in verbose natural language reasoning.

Our goal is to teach LVLMs to use interleaved formal-natural language reasoning. While prompt strategies seem straightforward, our evaluation shows that state-of-the-art LVLMs (e.g., OpenAI-o4-mini) frequently revert to natural language or create undefined operators instead of using geometry operators (Appendix D.3). This indicates LVLMs lack knowledge of geometric formal languages, likely because such documentation is absent from their training corpora. We address this data scarcity in the following sections.

## 3.2 Interleaved Formal-Natural CoT Data Synthesis

In this section, we curate high-quality data to teach LVLMs formal language reasoning. Our approach leverages advanced LVLMs that, while lacking specialized geometry formal language knowledge, demonstrate exceptional reasoning and instruction-following capabilities suitable for synthetic data generation. Our core strategy provides comprehensive operator definitions and detailed examples within prompts to guide LVLMs in combining reasoning with formal language. We formally define our synthesis setup using the following notations:

- `spec`: Detailed description of formal operands and operators and solution format requirements.
- `examples` $= \langle x_i, z_i, p_i \rangle_{i=1}^n$: Set of manually written example demonstrations showing interleaved formal-natural CoT reasoning using geometric symbols.
- $x$: Target question to be solved, containing the geometric problem statement and diagram.
- $z$: Synthetic interleaved formal-natural CoT reasoning trajectory generated by the state-of-the-art LVLM.
- $p$: Final formal program representing the executable solution.

The examples demonstrate a response style that first formalizes the problem using both problem variables and process variables, then explicitly cites relevant operators and theorems to derive the final solution. This approach aligns with the deliberative alignment approach in (Guan et al., 2024).

**Synthesis Objective.** Given the structured prompt containing formal description, demonstration, target question, and the ground-truth program (when available), our goal is to generate high-quality interleaved formal-natural reasoning $z$ that: 1) demonstrates step-by-step problem solving using geometric operators, (2) produces a valid formal program that can be executed by the geometric solver. To achieve this, we design two complementary synthesis strategies tailored to different dataset types, as illustrated in Figure 3. For datasets with only numerical solutions (e.g., UniGEO (Chen et al., 2022), Geo170k (Gao et al., 2025)), we utilize *forward synthesis* to generate both reasoning steps and formal program solutions from the given problems. For datasets with ground-truth formal program annotations (e.g., PGPS9k (Zhang et al., 2023a)), we employ *backward synthesis* to derive reasoning trajectories by backtracking from problems to their formal program solutions. The synthesis prompts are provided in Appendix D for reference.

**Forward Synthesis.** Given the structured prompt (`spec`, `examples`, $x$), the LVLM will generate $\widehat{y} = [\widehat{z}, \widehat{p}]$, where $\widehat{z}$ represents the model-generated interleaved formal-natural CoT reasoning trajectory and $\widehat{p}$ denotes the corresponding formal program solution. To filter out valid synthetic samples with correct reasoning chain $\widehat{z}$ and program $\widehat{p}$, we execute $\widehat{p}$ with a geometric solver and com-

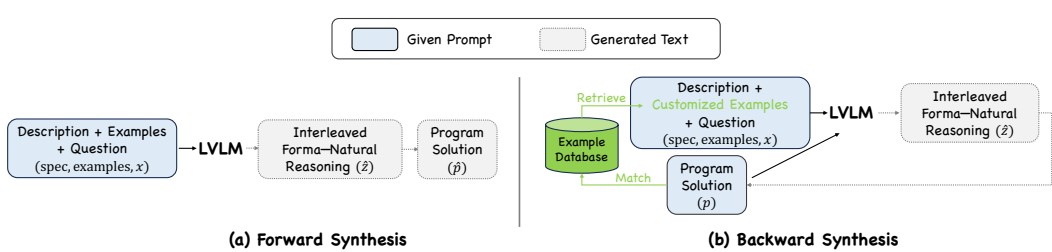

Figure 3: Overview of the interleaved formal-natural CoT data synthesis process.

pare its output to the ground-truth numerical solution. While flexible, forward synthesis achieves low success rates ($< 20\%$ for Qwen-2.5-VL-72B), highlighting the difficulty of establishing formal language reasoning through prompting alone. To address this, we leverage existing datasets with labeled programs (e.g., PGPS9K) as valuable hints for LVLMs, leading to our backward synthesis approach.

**Backward Synthesis.** We develop another backward synthesis method using the extended prompt $(\texttt{spec}, \texttt{examples}, x, p)$ that includes the correct program solution. The model generates the reasoning chain $\hat{z}$ that leads to the given $p$. We design a customized backward synthesis strategy to retrieve corresponding in-context learning examples from a manually written example database by matching the first operator in the solution program. This method further improves synthesis accuracy (shown in Table 4 later). The improvement stems from the model's prior knowledge of the correct program, enabling it to concentrate on reconstructing the reasoning path that links the question to the formal solution.

In total, we synthesize 11k interleaved formal-natural CoT reasoning samples, comprising 2.2k and 3.8k samples generated through forward data synthesis on the UniGEO and Geo170K training sets, and 5k samples produced via backward synthesis on the PGPS9K training set. Synthetic examples are provided in Appendix D.4.

### 3.3 TRAINING PROCEDURES

**Stage 1: Cold-Start Supervised Fine-tuning.** Using the synthetic reasoning data, we can conduct supervised fine-tuning of the LVLM to establish foundational interleaved formal-natural reasoning capabilities. The training objective is:

$$\min_{\theta} \mathbb{E}_{(x,z,p)\sim\mathcal{D}}[-\log \pi_\theta(z, p|x)], \tag{1}$$

where $\mathcal{D}$ denotes our synthetic dataset containing geometry problems paired with interleaved formal-natural reasoning chains and programs, and $\pi_\theta$ is the probability distribution of the LVLM.

This initial stage serves as a warm-up phase, equipping the model with two fundamental capabilities: (1) autoformalization of informal inputs into a format suitable for formal reasoning, and (2) basic comprehension of geometric formal language syntax. We empirically find that SFT alone is insufficient to achieve strong performance given the training samples and compute budget available (See Section 4.3). One possible explanation is that training with offline data constrains the model's behaviors. To address this limitation, we explore RL with online self-generated data to improve generalization performance.

**Stage 2: Solver-Integrated Reinforcement Learning.** After training on reasoning samples in the first stage, the model possesses effective exploration capability and basic formal language reasoning ability, though these abilities are not yet perfect. The subsequent reinforcement learning stage further refines these abilities through solver-guided optimization. In this phase, the model interacts with a geometric solver that provides verification results for the output program, enabling trial-and-error learning. The objective is as follows:

$$\max_{\theta} \mathbb{E}_{(z,p)\sim\pi_\theta(\cdot|x)}[r(z, p)], \tag{2}$$

where the reward function $r(z, p)$ is computed by executing the generated program $p$ through the geometric solver to obtain a numerical result for the target variables specified in the question, where

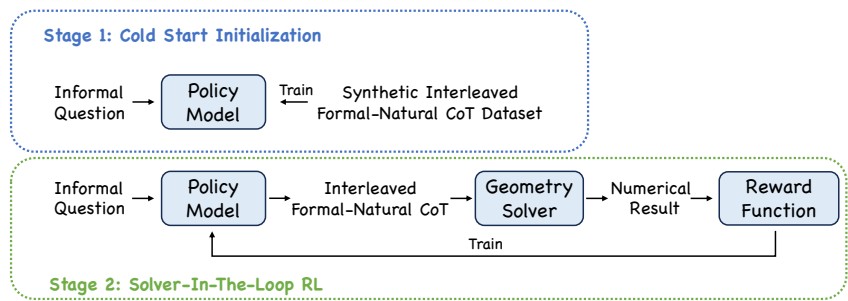

Figure 4: The two-stage training pipeline of our framework.

$r(r, p) = 1$ if the result matches the ground-truth solution, and 0 otherwise. We opt the GRPO algorithm (Shao et al., 2024) with a higher clip ratio (Yu et al., 2025).

## 4 EXPERIMENTS

### 4.1 EXPERIMENT SETUP

**Baselines.** We compare our method against three categories of baselines: 1) Specialist geometry systems for geometry calculation problems, comprising GeoX (Xia et al., 2024), NGS (Chen et al., 2021), Geoformer (Chen et al., 2022), and PGPSNet (Zhang et al., 2023a). 2) Open-Source General LVLMs, which include open source models Qwen2.5-VL-7B (Bai et al., 2025), Qwen2.5-VL-72B, InternVL3-8B-Instruct (Zhu et al., 2025), Vision-R1-7B (Huang et al., 2025), Kimi-VL-A3B-Instruct (Team et al., 2025), and Kimi-VL-A3B-Thinking. 3) State-of-the-art Closed-Source LVLMs including GPT-4o (2024-11-20) and Claude-3.7 Sonnet (2025-02-19).

**Datasets.** We assess geometric reasoning capabilities across four established benchmarks: (1) the test splits of PGPS9K and UniGEO, and (2) the plane geometry subsets from MathVista (Lu et al., 2024b) and MathVerse (Zhang et al., 2024) testmini splits, which provide cross-domain evaluation of mathematical visual reasoning.

**Evaluation Metrics.** Performance is measured by comparing numerical solutions against ground-truth answers, reporting the Pass@1 accuracy. To eliminate choice bias when evaluating on Math-Vista and MathVerse, we reformulate multiple-choice questions as free-form generation questions, ensuring fair assessment of the model's inherent reasoning ability.

### 4.2 MAIN RESULTS

**Comparison with Specialist Geometry Systems.** Figure 2 (left) demonstrates our method's performance on geometric problem solving compared to specialist geometry systems.[4] Our method achieves significantly better performance than existing geometry solvers, with gains of +16 points on PGPS9K and +18.3 points on UniGEO. Additionally, it is noteworthy that our method exhibits much higher flexibility by eliminating the need for pre-formalized questions and additional clauses. It can extract problem variables and formalize informal questions directly in the reasoning process. In contrast, specialist systems have limited adaptability due to their restrictive requirements: (1) GeoX, NGS, and Geoformer require problem variable formalization within the text question. (2) PGPSNet requires additional structured and semantic clauses beyond the text question and image diagram, which are typically unavailable in standard geometry problems.

**Comparison with Frontier LVLMs.** Table 2 demonstrates our method's performance compared with existing frontier LVLMs (mostly released this year). Among the evaluated baselines, we notice that Qwen2.5-VL-72B-Instruct achieves the strongest baseline results. Compared with it, our method achieves consistent superiority across evaluated benchmarks, achieving significant performance gains of +15.2 points on PGPS9K and +4.8 points on UniGEO. These improvements highlight our approach's superiority in multiple domains of geometry solving problems.

---

[4]The baseline scores are sourced from (Xia et al., 2024).

Table 2: Performance comparison with fountier LVLMs.

| Model | Params | PGPS | UniGeo | MathVista | MathVerse |
|---|---|---|---|---|---|
| GPT-4o (2024-11-20) | - | 50.9 | 43.9 | 47.1 | 43.3 |
| Claude-3.7 Sonnet (2025-02-19) | - | 53.7 | 47.2 | 52.4 | 47.2 |
| Qwen2.5-VL-7B-Instruct (Bai et al., 2025) | 7B | 39.4 | 51.5 | 52.4 | 39.9 |
| Qwen2.5-VL-72B-Instruct (Bai et al., 2025) | 72B | 53.3 | 67.9 | 63.0 | 52.1 |
| Vision-R1-7B (Huang et al., 2025) | 7B | 50.5 | 60.9 | 59.1 | 39.9 |
| InternVL3-8B-Instruct (Zhu et al., 2025) | 7B | 42.1 | 50.0 | 50.5 | 38.7 |
| Kimi-VL-A3B-Instruct (Team et al., 2025) | 16B | 37.0 | 42.2 | 42.8 | 36.4 |
| Kimi-VL-A3B-Thinking (Team et al., 2025) | 16B | 49.2 | 48.7 | 62.5 | 44.1 |
| **GF-Reasoner (Ours)** | 7B | **68.7** | **72.7** | **64.9** | **52.2** |

The integration of formal language also enhances token efficiency, enabling more concise and precise reasoning processes. As demonstrated in Figure 2 (right), among open-source models with fewer than 16B parameters, thinking models (e.g., Vision-R1-7B, Kimi-VL-A3B-Thinking) achieve higher accuracy at the cost of increased token consumption, while short-CoT models (e.g., Qwen2.5-VL-7B) prioritize token efficiency but sacrifice accuracy. Our interleaved formal-natural model breaks this trade-off by low token usage while simultaneously improving geometry problem-solving accuracy by 15% over the most effective baseline. This token efficiency gain stems from formal language's capacity to eliminate redundant natural language explanations.

### 4.3 ANALYSIS

In this section, we conduct ablation studies to systematically evaluate our framework's key components. Our analysis focuses on three critical aspects that contribute to the overall performance: (1) reasoning paradigm, particularly the benefits of bridging formal language with CoT reasoning in test-time scaling[5] and error reduction, (2) data synthesis strategies for training data preparation, (3) SFT and RL training recipes for establishing foundational knowledge and strategy refinement.

**Enhanced Inference-Time Performance through Bridging Formal Language with CoT.** We conduct a controlled study comparing bridging formal language with CoT reasoning against direct formal language prediction. We perform SFT on the Qwen-VL-2.5-7B-Instruct model using two carefully curated datasets derived from PGPS9K, matched in size and sample composition. The first dataset includes responses with step-by-step formal-language-interleaved reasoning trajectories, while the second contains only final formal program outputs. As shown in Figure 5, integrating formal language with CoT reasoning demonstrates superior Pass@K scaling, with the performance gap widening as the number of samples increases. Notably, on the PGPS9k dataset, CoT and non-CoT achieve comparable performance in Pass@1 evaluation, while CoT achieves a 7% relative improvement in Pass@8. This evidence suggests CoT's intermediate reasoning steps enable more effective exploration of the solution space during inference. Our finding aligns with (Prystawski et al., 2023), demonstrating that CoT enables the chaining of local knowledge to estimate relationships between variables not observed together during training.

**How Integrating Formal Language in Reasoning Reduces Errors?** To evaluate the advantages of integrating formal language in reasoning over traditional natural language reasoning, we categorize errors into four distinct types and evaluate how each type of error is reduced. The error type categorization follows existing works (Zhang et al., 2024; Lu et al., 2024b): 1) visual perception errors, 2) reasoning errors, 3) geometric knowledge errors, and 4) computation errors. Representative examples of each error type are shown in Appendix C.

To provide a comparative baseline for solving the geometry problems in natural language reasoning, we trained another natural language reasoning model using the same base model on the same dataset as our model with reinforcement learning. The reward function was calculated by comparing the extracted numerical solution from the generated response with the ground truth solution.

---

[5]Test-time scaling refers to leveraging additional computational resources during inference to enhance model performance (Snell et al., 2024).

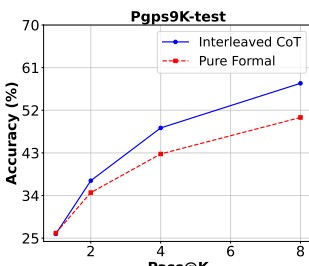 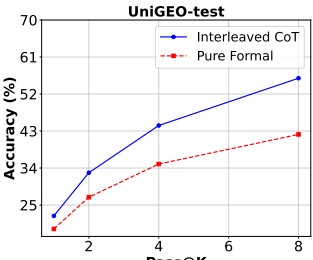 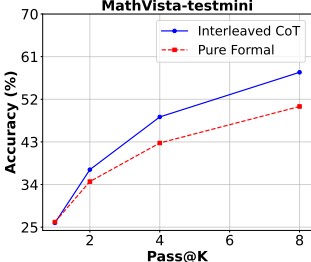

Figure 5: Pass@K performance comparison of supervised fine-tuned models trained with pure formal language versus interleaved formal-natural reasoning.

Table 3: Four types of errors in models using natural language versus interleaved formal-natural reasoning.

| Error Type | Natural Language | Interleaved Formal-Natural |
|---|---|---|
| Reasoning Error | 23.0% | **14.3%** |
| Geometry Knowledge Error | 12.3% | **10.3%** |
| Computation Error | 1.7% | **0.3%** |
| Visual Perception Error | **3.0%** | 8.0% |
| All | 40% | **32.9%** |

We conducted an error analysis on the same problem set (300 samples from PGPS9K-test) to highlight the differences between our interleaved formal-natural model and the natural language baseline model. The error analysis, presented in Table 3, demonstrates reductions in reasoning errors, geometry knowledge errors, and computation errors. Notably, interleaved formal-natural reasoning reduces reasoning errors by 8.7%. Besides, by offloading symbolic computation to an external solver, interleaved formal-natural reasoning reduces computation errors to nearly zero despite the already low baseline computation error rate.[6] For concrete examples of how formal reasoning eliminates computation and reasoning errors, refer to Appendix C. We can observe that interleaved formal-natural reasoning offers a more reliable and precise reasoning pathway than pure natural language.

**Customized Backward Synthesis for Improved Synthesis Accuracy.** Table 4 compares three synthesis strategies (forward, backward, and customized backward) using Qwen-VL-2.5-72B. We evaluate synthesis accuracy by extracting both the formal solution program and problem variables, then verifying them against ground truth numerical solutions using a geometry solver.

Our results demonstrate that backward synthesis yields a significant accuracy improvement (19.5% → 47.5%). This suggests that reconstructing the reasoning path from question to formal solution is more effective when the formal solution destination is known. Further performance gains (47.5% → 50.0%) are achieved by customizing in-context learning examples based on the target solution program, highlighting the importance of example-question alignment for efficient synthesis.

Table 4: Comparison of different data synthesis strategies.

| Method | Accuracy(%) |
|---|---|
| Forward | 19.5 |
| Backward | 47.5 |
| Customized Backward | **50.0** |

**Moderate SFT Preserves Reasoning Potential.** Previous studies in natural language reasoning (Li et al., 2025b; Zeng et al., 2025) suggest that excessive SFT may compromise response diversity, consequently limiting the model's reasoning capacity establishment in RL training. Our experiment results also verify this point. We perform an ablation study examining varying durations of cold start training (epochs) and their impact on subsequent RL performance. As illustrated in Figure 6, intensive fine-tuning (8 epochs, yellow color line) yields diminishing performance, showing sluggish performance gains during RL post-training. While moderate fine-tuning (2 epochs, red color

---

[6]The computation error in interleaved formal-natural reasoning is not zero because there is one sample where the model incorrectly calculated intermediate numerical results in natural language without using formal language.

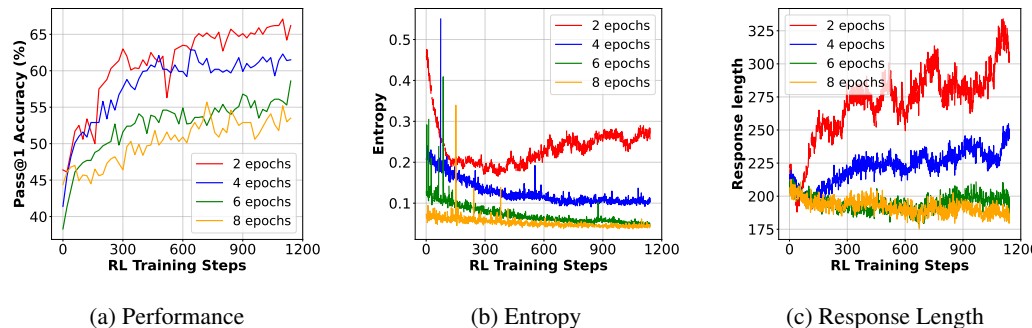

|                    |                    |                    |
|:------------------:|:------------------:|:------------------:|
| (a) Performance    | (b) Entropy        | (c) Response Length |

Figure 6: Performance, entropy, and response length on the PGPS9K test set during RL training varying training epochs of SFT initialization.

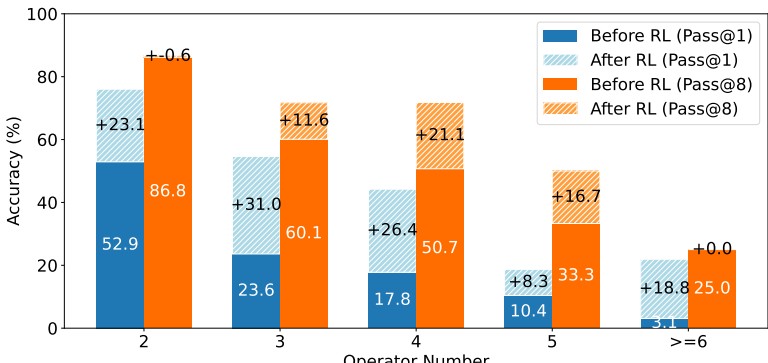

Figure 7: Performance on PGPS9K before/after RL training across different operator numbers.

line) achieves substantially better RL training efficiency. Additional entropy and response length measurements during RL training further confirm that the moderate cold start preserves response diversity, creating a more exploratory policy space that ultimately unlocks greater reasoning potential during RL post-training.

**RL Training Delivers Substantial Performance Gains Across Problem Complexities.** To investigate how RL training impacts problem-solving capability, we stratify the PGPS9k test problems into five levels by operator count (2, 3, 4, 5, and $\geq 6$), and report the Pass@K performance before and after RL training. As indicated in Figure 7, RL training demonstrates clear benefits: 1) RL enhances Pass@1 performance across all complexity levels, showing universal benefit in generating correct solutions on the first attempt. 2) RL delivers significant Pass@8 performance improvements for medium-complexity problems: +11.6% (3 ops), +21.1% (4 ops), and +16.7% (5 ops). While neither easy (operator=2) nor harder problems (operator $\geq 6$) show little Pass@8 improvement, suggesting RL primarily optimizes tasks where the base model has learnable but suboptimal strategies.

## 5 CONCLUSION

In this paper, we introduce a new hybrid reasoning paradigm for solving geometry problems, combining natural language with formal reasoning steps to leverage the complementary strengths of both approaches. To facilitate this framework, we curate an 11k-sample dataset featuring interleaved formal-natural CoT reasoning, including auto-formalization and natural-formal interleaved reasoning trajectories. Using this dataset, we investigate post-training procedures (SFT and RL) to train models in hybrid reasoning. Our results demonstrate that GF-Reasoner yields significant improvements in geometric problem-solving performance. We further provide comprehensive analysis of critical design choices, offering valuable insights for future research.

## 6 ETHICS AND REPRODUCIBILITY STATEMENT

This work focuses on advancing geometric problem-solving capabilities in LVLMs. The synthetic dataset generation process does not involve human subjects or sensitive data. Our approach aims to enhance mathematical education tools, which has positive societal implications.

In implementation details, we provide details about dataset construction, training frameworks, and hyper-parameters, which are sufficient to reproduce the experimental results. We will release the code, dataset, and models for research purposes to ensure full reproducibility.

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

## A  USE OF LLMS

In this work, LLMs were used to assist with writing refinement and polishing to improve the clarity and presentation of our findings. Additionally, LLMs were utilized as coding assistants to help implement certain components of our methodology, including data processing pipelines and evaluation scripts. The core algorithmic contributions, experimental design, and scientific insights are the original work of the authors.

## B  DISTINCTIONS FROM OTHER FRAMEWORKS

**Comparison with ReACT.** Our proposed method is conceptually similar to ReAct (Yao et al., 2022) at high-level in terms of interleaving reasoning and actions. However, there are fundamental differences in problem setting, technical challenges, and contributions.

- Inference-time prompting vs. Learning interleaved reasoning capabilities. ReAct is primarily an inference-time prompting framework that enables pre-trained models to alternate between reasoning and acting for general task solving. It assumes the model already possesses the necessary reasoning and action capabilities, and focuses on how to better orchestrate them through prompting. However, as shown in Table 7, directly prompting state-of-the-art models to perform interleaved formal-natural reasoning in geometry yields poor results. In contrast, our work addresses a fundamentally different challenge: teaching models to acquire the capability to perform interleaved formal-natural reasoning through post-training. Geometry problem solving requires models to learn domain-specific formal operators and how to properly interleave them with natural language reasoning—capabilities that do not exist in base models.

- Natural language actions vs. Formal symbolic operations. ReAct interleaves natural language reasoning with natural language actions (e.g., "Search[topic]" or API calls). Our approach interleaves natural language with formal symbolic operations that require: strict syntactic correctness; symbolic precision and mathematical rigor; domain-specific geometric operators (e.g., angle calculations, geometric constructions). This presents distinct technical challenges in data curation, training stability, and ensuring both reasoning coherence and formal correctness.

- Our work makes several contributions beyond applying an interleaving concept: 1) Curated training data: A interleaved formal-natural CoT dataset specifically designed for geometry, filling a gap in the community; 2) Systematic training strategies: Comprehensive study of SFT and RL approaches for learning interleaved reasoning; 3) Domain-specific insights: Analysis of how interleaving maintains exploration potential and reduces reasoning and computation errors in mathematical problem solving.

Thus, while ReAct demonstrates the value of interleaving at inference time for general tasks, our work shows how to effectively train models to perform interleaved formal-natural reasoning in specialized mathematical domains. These are complementary contributions addressing different aspects of the broader challenge.

**Comparison with Thinking with Images.** Recently, "Thinking with Images" has been introduced as a framework that leverages enhanced visual intelligence to solve problems through detailed image analysis, seamlessly integrating advanced reasoning with tools such as web search and automated image manipulations (e.g., zooming, cropping, flipping, and enhancement) to extract insights even from imperfect visual inputs (OpenAI, 2025). Our approach is orthogonal and complementary to this vision. While "Thinking with Images" emphasizes visual representations and planning in the image space, leveraging visual reasoning for geometric understanding. Our approach proposes a new reasoning paradigm in the language space that interleaves formal symbolic computation with natural language reasoning, aiming to achieve both mathematical precision (through formal language) and reasoning flexibility (through natural language). Therefore, these two frameworks are not conflicting but orthogonal, they focus on different representation spaces. An exciting future direction would be to combine both paradigms. We will explore this integration in future work.

Table 5: Progressive component analysis

| Config | Reasoning Paradigm | Data Synthesis | Style | Training Method | Pass@1 | Pass@8 |
|---|---|---|---|---|---|---|
| A | Pure Formal | Forward Only | - | SFT Only | 18.3 | 40.4 |
| B | Interleaved | Forward Only | Qwen | SFT Only | 20.8 | 52.6 |
| C | Interleaved | Bidirectional | Qwen | SFT Only | 31.2 | 61.1 |
| D | Interleaved | Bidirectional | o4-mini | SFT Only | 42.1 | 76.5 |
| **E (Ours)** | **Interleaved** | **Bidirectional** | **o4-mini** | **SFT + RL** | **68.7** | **80.7** |

Table 6: Reasoning paradigm comparison

| Reasoning Paradigm | PGPS9K | UniGEO |
|---|---|---|
| Pure Formal Language Reasoning | 51.8% | 55.2% |
| Pure Natural Language Reasoning | 60.0% | 68.7% |
| Interleaved Formal-Natural CoT | 68.7% | 72.7% |

Table 7: Performance on PGPS9k when given access to formal language description and demonstration examples.

| Method | Accuracy |
|---|---|
| GPT-4o | 6.4% |
| Claude-3.7 Sonnet | 16.2% |
| Qwen-2.5-72B-VL-Instruct | 14.0% |
| Qwen-2.5-7B-VL-Instruct | 0.1% |

## C ADDITIONAL RESULTS

**Progressive Component Analysis.** The ablation study in Table 5 reveals progressive performance improvements across configurations. Moving from pure formal language (Config A) to interleaved reasoning (Config B) provides +12.2% Pass@8 gain, showing that integrating formal language with CoT enables more effective exploration of the solution space. Bidirectional synthesis (Config C) dramatically outperforms forward-only synthesis, achieving +10.4% improvement on Pass@1, due to more synthetic data obtained by leveraging ground-truth programs as synthesis hints. Transfer from Qwen to o4-mini reasoning style for data synthesis (Config D) delivers a +10.9% Pass@1 boost, highlighting the importance of advanced reasoning capabilities for generating coherent interleaved formal-natural chains. Adding solver-in-the-loop RL, our complete approach (Config E) achieves 68.7% Pass@1, representing a remarkable +26.6% improvement over Config D, demonstrating that RL enables the model to learn from solver feedback and refine its formal reasoning strategies through iterative improvement.

**Reasoning Paradigm Comparison.** In Section 4.3, we showed that interleaved CoT offers better exploration than pure formal solutions and reduces reasoning, knowledge, and computational errors compared to pure natural language. We further provide quantitative results in Table 6. Our comparisons use the same dataset and training procedures (SFT+RL) for all three paradigms. The result confirms that interleaved CoT consistently outperforms both pure approaches, demonstrating the benefits of interleaving in improving performance.

**Performance Gains Beyond Tool Access.** To isolate the performance gain given access to tools, we provided baseline models (GPT-4o, Claude-3.7 Sonnet, Qwen-2.5-VL series) with identical formal language description and demonstration examples (used in our training data synthesis) via detailed prompting. This ensures they also have "tool access". As shown in Table 7, despite having explicit access to formal specifications, these strong foundation models still struggle to generate valid interleaved CoT the solver can execute, resulting in low accuracy. This result demonstrates that our performance gains cannot be attributed to given access to tools alone, but stem from teaching models to internalize solver knowledge and reasoning patterns.

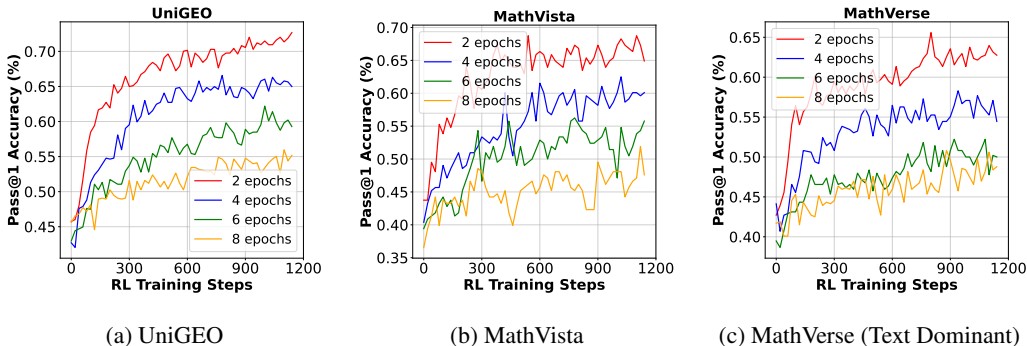

(a) UniGEO        (b) MathVista        (c) MathVerse (Text Dominant)

Figure 8: Performance on the UniGEO, MathVista, and MathVerse (Text Dominant) test set during RL training varying training epochs of SFT initialization.

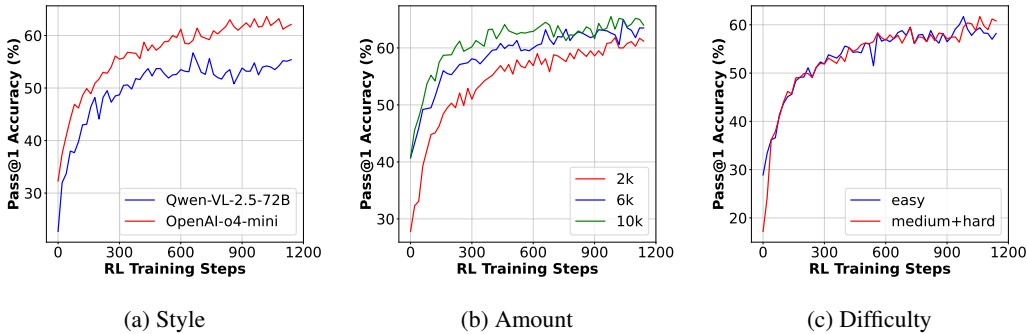

(a) Style        (b) Amount        (c) Difficulty

Figure 9: Performance on the PGPS9K test set during RL training varying SFT initialization dataset, including CoT trajectory style, dataset amount, and difficulty coverage.

**SFT Epoch Influence on More Benchmarks.** In Figure 6 in the main text, we observe that moderate SFT preserves reasoning potential on PGPS9K dataset. This phenomenon can also be observed on other benchmarks. We have evaluated checkpoints across multiple benchmarks (UniGEO, MathVista, and MathVerse). As shown in Figure 8, the results demonstrate that while SFT is important to build fundamental capacity, heavy SFT may compromise response diversity, consequently limiting the model's reasoning capacity establishment in RL training.

**Impact of SFT Training Data Characteristics.** The quality and composition of interleaved formal-natural CoT data during SFT training directly influence the base model's mastery in the formal language for geometry problem solving, and consequently affect reasoning trajectory refinement in RL training. We systematically investigate three dimensions of interleaved formal-natural data, analyzing several key variants:

- **CoT Trajectory Style**: We compare the base models trained on interleaved formal-natural reasoning trajectories generated by OpenAI-o4-mini and Qwen2.5-VL-72B, respectively, both are synthesized based on PGPS9K training set with an equal amount (3k each) and identical problem coverage.

- **Dataset Amount**: With the OpenAI-o4-mini CoT style, we experiment by varying the number of training examples used, specifically using 2k, 6k, and 10k randomly sampled examples as SFT training data, respectively.

- **Difficulty Coverage**: According to the number of geometry operators used, we categorize the training dataset into two difficulty levels, including easy ($\leq 3$ operators) and medium/hard ($\geq 4$ operators). We create equal-sized subsets for each difficulty level (2k each) and train the base model on the two subsets.

We perform the ablation study varying each of the three dimensions independently to isolate their impact on model performance. Results are presented in Figure 9.

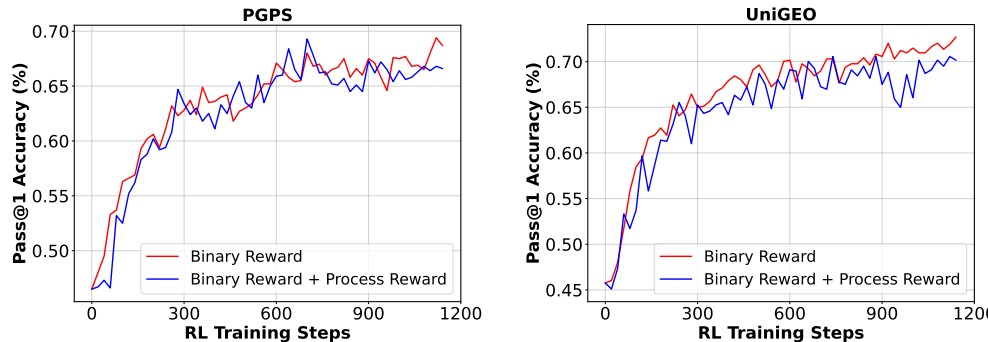

Figure 10: The performance during RL training with and without process reward.

Table 8: SFT initialization results on a larger base model.

| Base Model | Pass@1 | Pass@2 | Pass@4 | Pass@8 |
|---|---|---|---|---|
| Qwen-2.5-VL-7B | 42.2 | 55.3 | 67.2 | 77.1 |
| Qwen-2.5-VL-32B | 56.5 | 67.9 | 77.3 | 84.6 |

The results in Figure 9(a) demonstrate that RL training initialized with OpenAI-o4-mini-style CoT data consistently outperforms training based on Qwen-VL-2.5-72B-style data. To investigate this discrepancy, we analyze the reasoning trajectories of both models. We observe that OpenAI-o4-mini generates more logically consistent and formally precise reasoning steps, leading to better policy initialization for RL. Additionally, the results in Figure 9(b) indicate that increasing the amount of SFT initialization data improves the final RL performance, but the gains diminish beyond a certain threshold (e.g., 6k samples). This suggests that a relatively small but well-curated dataset is sufficient for the model to learn the fundamental grammar and usage of geometric formal language during SFT. Additional data provides only marginal improvements. Interestingly, from Figure 9(c), we could also observe that the difficulty level of SFT data (easy vs. medium/hard) does not significantly affect final RL performance. Although models trained on medium/hard CoT data initially underperform those trained on easy data, both converge to similar accuracy after RL training. This implies that SFT primarily serves to teach foundational knowledge of formal language, which can be acquired from either easy or hard problems, while RL compensates for higher-level reasoning gaps.

**On the Design of the Reward Function.** We investigated extending the binary final-answer reward with partial credit for intermediate steps. Specifically, we defined a process reward that checks the *grammatical validity* of the generated formal program (i.e., reward=1 if the program is syntactically correct). As shown in Figure 10, adding this process reward during RL training dose not yield gains compared to using the binary outcome reward alone. We hypothesize that this is because the RL training's multi-trajectory sampling already provides implicit partial credit: correct intermediate steps occur more frequently in successful trajectories, allowing the model to statistically learn which reasoning patterns lead to correct outcomes.

**Training on a Larger Base Model.** To investigate whether this framework is scalable to larger models, we conduct additional experiments using Qwen-2.5-VL-32B as our base model. The results are shown in Table 8, it can be observed that using a larger base model, the Pass@8 performance increases from 77.1 to 84.6. This demonstrates that our training framework effectively scales with model capacity.

**Comparison with Recent Frontier Models.** We evaluate our approach against several recently released frontier large multimodal models, as summarized in Table 9. The compared models include Qwen3-VL-8B-Instruct, GLM-4.5V, Gemini 2.5 Pro, GPT-5, and the "Thinking with Images" model (e.g., OpenAI o4-mini). We train two variants of our model: one based on Qwen2.5-VL-7B-Instruct and the other on Qwen2.5-VL-32B-Instruct. As shown in the table, our 32B model achieves competitive performance with significantly larger commercial models (e.g., GLM-4.5V with 106B parameters, and larger frontier models like Gemini 2.5 Pro, GPT-5, and o4-mini), demonstrating the

Table 9: Comparison with recent frontier models on the PGPS9K test set.

| Methods | Pass@1 Accuracy |
|---|---|
| Qwen2.5-VL-7B-Instruct | 39.4% |
| Qwen2.5-VL-32B-Instruct | 50.0% |
| Qwen3-VL-8B-Instruct | 70.9% |
| GLM-4.5V | 77.9% |
| Gemini 2.5 Pro | 83.3% |
| GPT-5 | **85.8%** |
| OpenAI-o3 (Thinking with Images) | 77.2% |
| Ours (Qwen2.5-VL-7B-Instruct) | 68.7% |
| Ours (Qwen2.5-VL-32B-Instruct) | 82.1% |

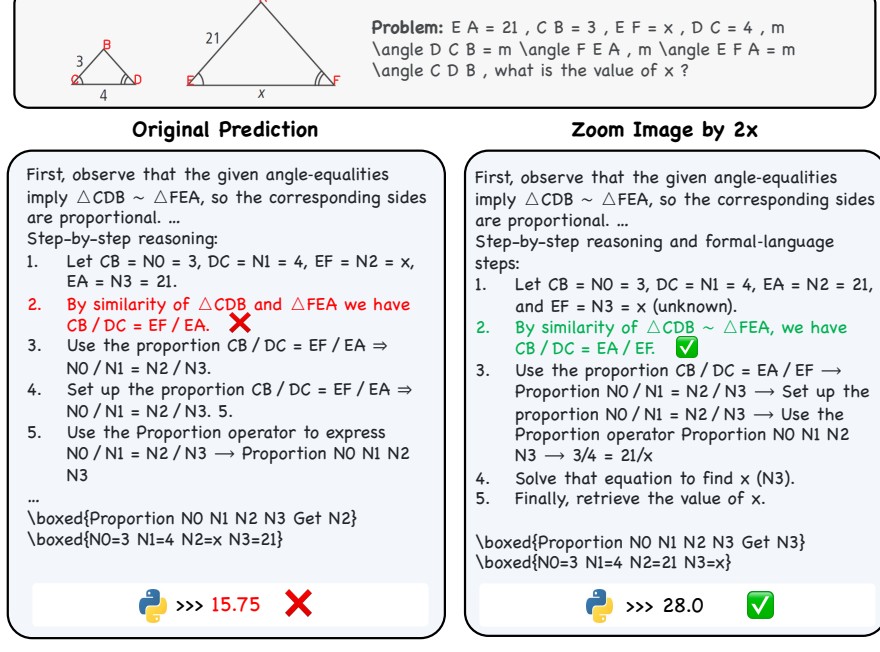

Figure 11: A case study on improving visual perception through image zooming.

effectiveness of our approach. It can also be noticed that our training framework scales effectively with model capacity: scaling from 7B to 32B yields approximately 14% performance improvement, suggesting that further gains are possible with larger base models. Our framework provides a scalable and efficient approach that can be applied to larger models as computational resources permit.

**Zooming Images for Better Perception.** We have tested integrating a zoom tool and found it effective. It provides higher-resolution details, which improves visual perception. On PGPS9K, this raised accuracy from 68.7% to 69.3% and reduced the visual perception error from 8.0% to 6.0%. A supporting case study is provided in Figure 11, where applying a 2x zoom enables the model to obtain a clearer view, leading to the correct identification of corresponding sides in a pair of similar triangles. These results suggest that incorporating image processing tools can aid visual reasoning and improve geometric understanding, thereby complementing the current framework.

**Error Analysis: Comparing Interleaved Formal-Natural vs. Natural Language Reasoning.** In Section 4.3, we categorize geometry problem-solving errors into four types: visual perception errors, reasoning errors, geometric knowledge errors, and computation errors.This error taxonomy is adapted from established frameworks in geometry problem-solving research (Zhang et al., 2024; Fu et al., 2025). The illustration of the four types of errors is described below:

- Visual perception errors: The model failed to either extract salient features from the image or associate the textual information with its corresponding geometric shapes.

- Reasoning errors: The model failed to generate a valid solution, demonstrating either a lack of logical coherence and internal consistency or a failure in procedural execution by skipping necessary operations.

- Geometric knowledge errors: The model's error was either in misapplying a geometric concept or in failing to distinguish between the definitions of two theorems.

- Computation errors: The model failed to calculate the numerical results, leading to errors in calculations involving arithmetic, square roots, and proportions.

We classify the wrong solution into one of the four types of errors based on the first error in the response. As follows, we illustrate representative cases for all four error types.

For annotation, two PhD students and one master student in Computer Science are involved, all with expertise in geometry problem solving and LVLMs. Each annotator independently labeled 300 test cases. Disagreements were resolved through discussion to reach consensus. The initial annotation achieved 91% inter-annotator agreement.

| #1 Visual Perception Error |
| --- |

**Problem**
E A parallel D B , E D = 4 , C B = 12 , A B = 5 , C D = x , find x .

**Image**

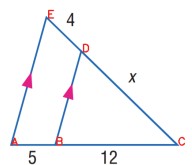

**Model's Response**

| Response | ...we need to use the properties of similar triangles. Since EA ∥ DB, △ AED and △ ABC are similar by the AA (Angle-Angle) similarity criterion. |
| --- | --- |
| Error | △ AED and △ ABC are not triangle. |

**Correct Solution**

| Thought | Since line segment EA is parallel to DB, it follows that triangle CEA is similar to triangle CDB (△ CEA ∼ △ CDB). |
| --- | --- |
| Answer | x = 9.6 |

| #2 Reasoning Error |
| --- |

**Problem**
m∠AXY = 22, m∠ AYX = 2x+2, and m∠ AZW = 3x-23, what is m∠ W?

**Image**

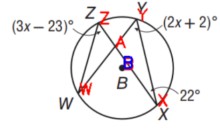

**Model's Response**

| Response | ...If we assume $x$=23 (a common assumption for such problems without additional constraints), then... |
| --- | --- |
| Error | The model couldn't solve for $x$ geometrically, so it instead assumed a "common" value of $x$=23, a step not based on the problem's logic. |

**Correct Solution**

| Thought | ∠W and ∠AXY are both subtended by the same arc ZY. According to the Inscribed Angle Theorem, ∠W = ∠AXY = 22°. |
| --- | --- |
| Answer | m∠W = 22° |

## #3 Geometric Knowledge Error

**Problem**

RQ = 6, QT = x, SR = 14, SQ = 18, and m∠ QRT = m∠ SRT, find x.

**Image**

**Model's Response**

| | |
|---|---|
| **Response** | ...We are given that angles (∠QRT = ∠SRT). This implies that △QTR and △SRT are similar by the Angle-Angle (AA) similarity criterion.... |
| **Error** | It incorrectly concludes that △QTR and △SRT are similar based on having only one pair of equal angles (∠QRT = ∠SRT). |

**Correct Solution**

| | |
|---|---|
| **Thought** | Since ∠ QRT = ∠ SRT, the segment RT bisects the ∠ QSR in △ QSR. According to the Angle Bisector Theorem, QR/QT = RS/TS. |
| **Answer** | x = 5.4 |

## #4 Computation Error

**Problem**

angle CDA = 70 , angle BAD = 110 , angle ABC = 3x-5 , angle DCB = 4x+10 , what is angle C ?

**Image**

**Model's Response**

| | |
|---|---|
| **Response** | To ... angle CDA + angle BAD + angle ABC + angle DCB = 360°. Substitute the given values: 70° + 110° + (3x - 5) + (4x + 10) = 360° ...Simplify the equation: 175° + 7x = 360°... |
| **Error** | Calculation mistake: 70° + 110° + (3x - 5) + (4x + 10) = 360 was incorrectly simplified as 175° + 7x = 360°. |

**Correct Solution**

| | |
|---|---|
| **Thought** | The sum of the interior angles of a quadrilateral is 360°. Thus, 70 + 110 + (3x - 5) + (4x + 10) = 360 ⟹ 185 + 7x = 360 ⟹ x = 25 |
| **Answer** | m∠ C = 4x + 10 = 110 |

Below, we demonstrate how our interleaved formal-natural reasoning model reduces computation error compared to the natural language reasoning baseline. This baseline model is trained with RL using the same base model and training dataset as our approach. The natural language model uses premature approximations in mathematical functions, leading to precision errors in the final result, while our model maintains exact symbolic computation throughout the reasoning process and only performs numerical evaluation at the final step through the geometric solver, thereby preserving mathematical precision and avoiding cumulative rounding errors.

---

**Computation Error Case**

**Question:**
$ED = 7, BE = EG = FG = CF = AC = AB, m\angle ABE = m\angle GEB = m\angle EGF = m\angle GFC = m\angle FCA = m\angle CAB$, find the area of the shaded region.

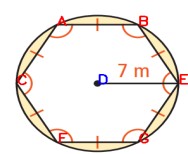

---

**Natural Language Solution**

**Model's Response:**
To find the area of the shaded region, we first need to understand that the given hexagon is inscribed in a circle with radius $ED = 7$ (i.e., $r = 7$). Since the hexagon is regular, each internal angle is $120°$ (since $360°/6 = 60°$, and each exterior angle is $60°$).
$\cdots$

The area $A_{\text{triangle}}$ of an equilateral triangle with side length $r$ (here $r = 7$) is:
Area of equilateral triangle $= \frac{\sqrt{3}}{4} \cdot r^2$
Substitute $r = 7$:
Area of one equilateral triangle $=$
$\frac{49\sqrt{3}}{4} \approx \frac{49 \cdot 1.732}{4} \approx \frac{84.784}{4} \approx 21.196$
Hence, area of hexagon $A_{\text{hexagon}}$ :
$A_{\text{hexagon}} = 6 \cdot 21.196 \approx 127.176$
Next, area $A_{\text{circle}}$ :
$A_{\text{circle}} = \pi \cdot r^2 = \pi \cdot 7^2 = 49\pi \approx 49 \cdot 3.14159 \approx 153.94$
Shaded area $A_{\text{shaded}}$ :
$A_{\text{shaded}} \approx 153.94 - 127.176 \approx 26.764$
Hence, the area of the shaded region is approximately:

$$\boxed{26.764}$$

- - - - - - - - - - - - - - - - -

**Answer:**
26.764 (Correct: 26.632) ✗

**Error:**
Premature use of approximations (e.g., $\sqrt{3} \approx 1.732$) led to a precision error in the final result.

---

**Interleaved Formal-Natural Solution**

**Model's Response:**
Step-by-step reasoning:
1. Let the circle's radius (and the circumradius of the hexagon) be
$N_0 = DE = 7$.
2. Compute the area of the circle with radius $N_0$.
Use `Circle_R_Area` to get the circle's area:
`Circle_R_Area` $N_0 \, V_0 \to V_0 = \pi \cdot N_0^2$
3. Compute the area of the regular hexagon with circumradius $N_0$.
Use `RNgon_L_Area` for a regular 6-gon of circumradius $N_0$:
`RNgon_L_Area` C6 $N_0 \, V_1 \to V_1 =$ area of the hexagon
4. The shaded region = circle area minus hexagon area.
So we take the circle area $V_0$ minus the hexagon area $V_1$.
Use `Sum` to express
$V_1 + V_2 = V_0 \to V_2$ is the shaded area.
Hence
`Sum` $V_1 \, V_2 \, V_0 \to V_1 + V_2 = V_0 \to V_2 = V_0 - V_1$
5. Extract the shaded-region area
Get $V_2$

```
Circle_R_Area N0 V0
```
```
RNgon_L_Area C6 N0 V1
```
```
Sum V1 V2 V0 Get V2
```
```
N0 = 7
```

- - - - - - - - - - - - - - - - -

**Answer:**
The solver executes the program and outputs 26.632. ✔

Below, we compare interleaved formal-natural reasoning model to the natural language baseline in reducing reasoning errors. The natural language model exhibits logical inconsistencies, makes unjustified assumptions (e.g., arbitrarily setting x=13), and falls into infinite reasoning loops when encountering contradictory results. In contrast, our interleaved formal-natural approach enforces structured reasoning through geometric operators, maintains logical consistency, and leverages external solver verification to prevent arbitrary assumptions and ensure mathematically sound conclusions.

---

### Reasoning Error Case

**Question:**
In $\odot B$, $m\angle LBM = 3x$ and $m\angle LBQ = 4x + 61$. $MQ \perp KB$ at $B$, what is the measure of $\angle PBQ$ ?

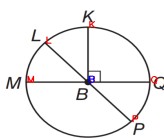

---

### Natural Language Solution

**Model's Response:**
Since $MQ \perp KB$ at $B$, $KB$ is the perpendicular bisector of $LM$ at $B$....$m\angle LBM = 3x$ and $m\angle LBQ = 4x + 61$... Hence, we can write the equation: $m\angle LBM = m\angle QBM$
Using the equation $3x = 4x + 61 - 3x$:
$x = -61$
Since $x$ cannot be negative, let us use the properties of circle :
m $\angle LBQ = 180 - 3x$.
Using $\angle LBQ = 4x + 61 = 4x + 61$.
$\angle PBQ = 90 - (4x + 61)$.
Using :
m $\angle LBQ = 4x + 61 = 4x + 61$.
...
$\angle PBQ = 4x + 61 = 4 * 13 + 61 = 4 * 13 = 52$.
$\angle PBQ = \angle QBM = 4x + 61 = 52$.
$\angle PBQ = \angle QBM = \angle PBQ = \angle PBQ = \angle PBQ = ...$

**Answer:**
52 (Correct: 51) ✗

**Error:**
The reasoning process involves random guessing, incorrectly assumes the value of x to be 13, and ultimately falls into an infinite loop.

### Interleaved Formal-Natural Solution

**Model's Response:**
Step-by-step reasoning:
1. Let
$N_0 = m\angle LBM = 3x$
$N_1 = m\angle LBQ = 4x + 61$
2. Since $\angle LBM$ and $\angle LBQ$ are supplementary on a straight line,
$N_0 + N_1 = 180$.
Use `Sum` $N_0$ $N_1$ `C180` $\to N_0 + N_1 = 180$
(3x + (4x + 61) = 180)
3. $\angle PBQ$ and $\angle LBQ$ are supplementary on the straight line through the diameter, so
$m\angle PBQ = 180 - m\angle LBQ$.
Hence
$V_0 = 180 - N_1$.
Use `Sum` $N_1$ $V_0$ `C180` $\to N_1 + V_0 = 180$
$\to V_0$ is $m\angle PBQ$
4. Extract the value of $\angle PBQ$
`Get` $V_0$

```
Sum N0 N1 C180
```
```
Sum N1 V0 C180 Get V0
```
```
N0 = 3*x N1 = 4*x + 61
```

---

**Answer:**
The solver executes the program and outputs 51. ✔

## D   DATASET DETAILS

We provide forward and backward synthetic prompts for generating Interleaved Formal-Natural CoT data samples in Section D.1 and Section D.2. Section D.3 provides a case study on directly prompting state-of-the-art LVLMs with human demonstrations. The results show that they lack prior knowledge of geometric formal language. Finally, we showcase Interleaved Formal-Natural CoT examples from our dataset in Section D.4.

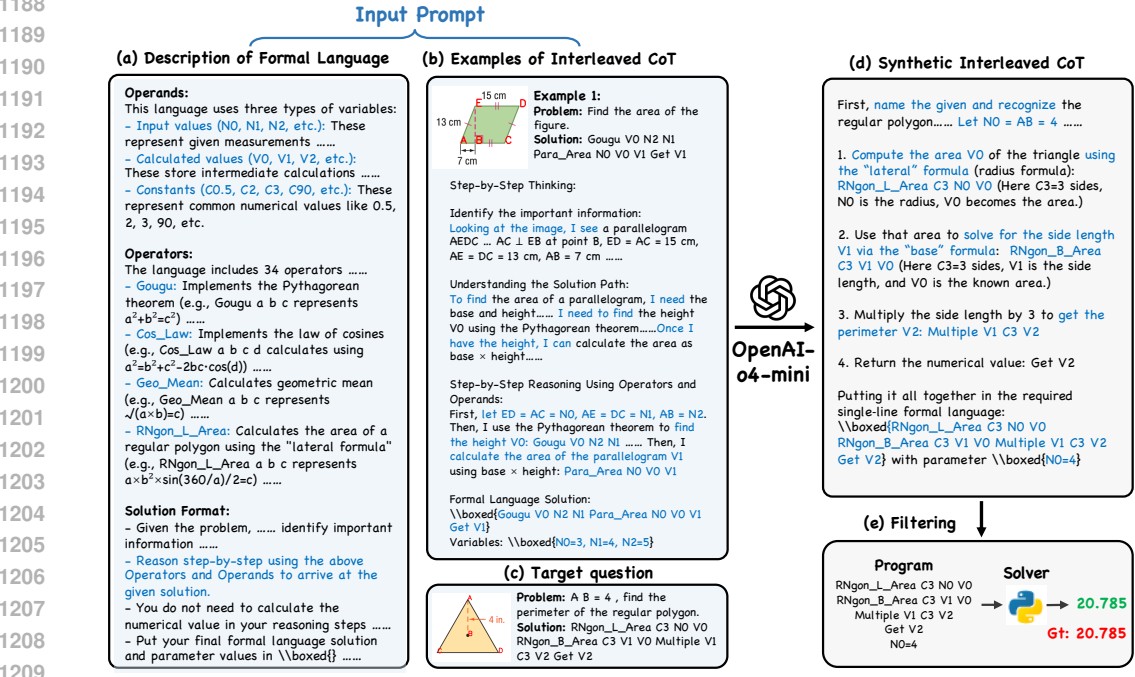

Figure 12: Detailed explanation of the backward synthesis process of interleaved formal-natural CoT data.

## D.1 FORWARD SYNTHETIC PROMPT

---

### Forward Synthesis Prompt

You are a geometry problem expert. You have access to a solver with the following formal language. The following prompt teaches you how to use this language through examples and explanations.

#### FORMAL LANGUAGE SPECIFICATION

##### OPERANDS

This language uses three types of variables:

- Input values (N0, N1, N2, etc.): These represent given measurements, lengths, angles, or other known values in a problem. The numbering must start from N0 and increment by 1.

- Calculated values (V0, V1, V2, etc.): These store intermediate calculations or final results. The numbering must start from V0 and increment by 1.

- Constants (C0.5, C2, C3, C90, etc.): These represent common numerical values like 0.5, 2, 3, 90, etc.

##### OPERATORS

The language includes 34 operators that represent geometric relations and calculations:

##### BASIC MATHEMATICAL OPERATIONS

- Get: Retrieves the numerical value of a variable (e.g., Get V0 returns the value stored in V0)

- Sum: Performs addition of multiple terms (e.g., Sum a b c d represents $a + b + c = d$)

---

- `Multiple`: Performs multiplication of multiple terms (e.g., `Multiple a b c d` represents $a \times b \times c = d$)
- `Equal`: Sets two expressions equal (e.g., `Equal a b` represents $a = b$)

TRIANGLE OPERATIONS

- `Gougu`: Implements the Pythagorean theorem (e.g., `Gougu a b c` represents $a^2 + b^2 = c^2$)
  - 'a' is the first leg of the right-angled triangle
  - 'b' is the second leg of the right-angled triangle
  - 'c' is the hypotenuse opposite the right angle
  ...

TRIGONOMETRIC OPERATIONS

- `Gsin`: Implements sine relation (e.g., `Gsin a b c` represents $\sin(c) = \frac{a}{b}$)
  - 'a' is the opposite side length
  - 'b' is the hypotenuse length
  - 'c' is the angle in degrees
  ...

QUADRILATERAL OPERATIONS

- `Para_Area`: Calculates parallelogram area (e.g., `Para_Area a b c` represents $a \times b = c$)
  - 'a' is the base length
  - 'b' is the height perpendicular to the base
  - 'c' is the resulting area
  ...

CIRCLE OPERATIONS

- `Circle_R_Circum`: Calculates circle circumference from radius (e.g., `Circle_R_Circum a b` represents $2\pi \times a = b$)
  - 'a' is the radius length
  - 'b' is the resulting circumference
  - Optional format: `Circle_R_Circum a b c` represents $2\pi \times a \times \frac{b}{360} = c$, where 'b' is the central angle in degrees and 'c' is the arc length.
  ...

OTHER GEOMETRIC RELATIONS

- `Geo_Mean`: Calculates geometric mean (e.g., `Geo_Mean a b c` represents $\sqrt{a \times b} = c$)
  - 'a' is the first value
  - 'b' is the second value
  - 'c' is the resulting geometric mean
  - This operator has numerous applications in geometric problems:
    * Tangent-Secant Power Theorem: `Geo_Mean PA PB PT` → PT is the tangent length when a point P outside the circle has a secant with segments PA and PB, where PT = $\sqrt{PA \times PB}$.
    * Altitude Rule (Right Triangle Projection): `Geo_Mean p q h` → In a right triangle, the altitude h to the hypotenuse is the geometric mean of the two segments p and q it creates on the hypotenuse, where h = $\sqrt{p \times q}$.
    * Leg Projection Rule: `Geo_Mean p c a` → A leg a of a right triangle is the geometric mean of its projection p on the hypotenuse and the hypotenuse c itself, where a = $\sqrt{p \times c}$.
  ...

## SOLUTION FORMAT

- Given the problem, you should first understand it and analyze the image to identify important information that can lead to the solution.

- Reason step-by-step using the above Operators and Operands to arrive at the final solution. You must explicitly state the Operator and Operand you used in your reasoning steps. You may use multiple sequences of Operators and Operands.

- You do not need to calculate the numerical value in your reasoning steps since the solver with your formal language answer can do this.

- Put your final formal language solution and parameter values in \boxed{} separately, both on a single line. Ensure that the content inside \boxed{} is space-delimited and compatible with Python's eval() for direct parsing by the solver.

## EXAMPLES

### EXAMPLE 1

**Problem:** Find the area of the figure.
**Image:**

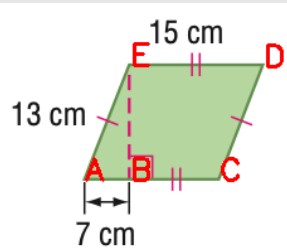

**Step-by-Step Thinking:** Looking at the image, I see a parallelogram AEDC with various measurements. From the image and problem statement: - $AC \perp EB$ at point $B$ - $ED = AC = 15$ cm - $AE = DC = 13$ cm - $AB = 7$ cm
I need to find the area of the parallelogram AEDC. Let me solve this by using geometric principles and operations from the formal language:

1. Identify the important information:
    - $AC \perp EB$ at point $B$
    - $ED = AC = 15$ cm
    - $AE = DC = 13$ cm
    - $AB = 7$ cm

2. Understanding the Solution Path:
    - To find the area of a parallelogram, I need the base and height.
    - I can use $ED = 15$ cm as the base.
    - For the height, I need the perpendicular distance from $ED$ to $AC$.
    - The problem states $AC \perp EB$ at $B$, which means $AC$ is perpendicular to $EB$.
    - I need to find the height $V0$ using the Pythagorean theorem with $AB$ and the distance from $A$ to the line $ED$.
    - Given $AB = 7$ cm and the distance from $A$ to $E$ is 13 cm, I can use the Pythagorean theorem to find the height.
    - Once I have the height $V0$, I can calculate the area as base $\times$ height.

3. Step-by-Step Reasoning Using Operators and Operands:
    - First, let $ED = AC = $ N0, $AE = DC = $ N1, $AB = $ N2.
    - Then, I use the Pythagorean theorem to find the height V0:

```
Gougu V0 N2 N1
```

This solves for `V0` where `V0`$^2$ + `N2`$^2$ = `N1`$^2$

- Then, I calculate the area of the parallelogram `V1` using base × height:

```
Para_Area N0 V0 V1
```

This computes `V1` = `N0` × `V0`

- Finally, I get the result:

```
Get V1
```

4. Formal Language Solution:

```
Gougu V0 N2 N1 Para_Area N0 V0 V1 Get V1
```

with parameter values:

```
N0=15, N1=13, N2=7
```

EXAMPLE 2

...

## YOUR TASK

**Problem:** {{Target problem}}
**Image:** {{Image diagram of the target problem}}
**Step-by-Step Thinking:**

### D.2 BACKWARD SYNTHESIS PROMPT

Customized Backward Synthesis Prompt for `Chord2_Ang_v0`

You are a geometry problem expert. You have access to a solver with the following formal language. The following prompt teaches you how to use this language through examples and explanations.

## FORMAL LANGUAGE SPECIFICATION

...

⟨Same with the forward synthesis prompt⟩

...

## EXAMPLES

EXAMPLE 1
**Problem:** Find $m\angle F$.
**Image:**

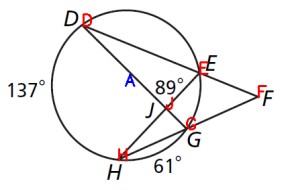

**Solution:** `Sum V0 N0 C180 Chord2_Ang V0 V1 N1 TanSec_Ang V2 V1 N1 Get V2`

**Step-by-Step Thinking:** Looking at the circle diagram, I can see several points ($D$, $E$, $F$, $G$, $H$, $J$) with some angles marked:

- $m\angle DJE = 89°$
- $m\angle HG$ (arc $HG$) = $61°$

- $m\angle DH$ (arc $DH$) $= 137°$
- I need to find $m\angle F$

The task is to find angle $F$. Let me solve this by using geometric principles and operations from the formal language:

1. Identify the Given Information:

- $m\angle DJE = 89°$
- $m\angle HG$ (arc $HG$) $= 61°$
- $m\angle DH$ (arc $DH$) $= 137°$
- Point $J$ is inside the circle
- Point $F$ is outside the circle
- Need to find $m\angle F$

2. Understanding the Solution Path:

- First, I notice that points $D$, $H$, $G$, and $E$ lie on the circle. Point $J$ is inside the circle, while point $F$ is outside the circle.
- Since $F$ is outside the circle with secant lines $FD$ and $FH$, I can use the `TanSec_Ang` theorem
- To apply this theorem, I need to find the intercepted arcs
- To find one of the arcs, I'll use the `Chord2_Ang` operator and properties of angles inside a circle

3. Step-by-Step Reasoning Using Operators and Operands:

- I'll first calculate the supplementary angle to $\angle DJE$, let $\angle DJE = N0$ and $\angle EJG = V0$:
  `Sum V0 N0 C180`
  (This calculates $V0 = 180° - N0 = 180° - 89° = 91°$)
- Since $J$ is a point inside the circle, I can use the `Chord2_Ang` theorem to calculate the arc $EG$, let arc $DH = N1$ and arc $EG = V1$:
  `Chord2_Ang V0 V1 N1`
  (This relationship comes from the inscribed angle theorem for intersecting chords)
- Finally, since $F$ is outside the circle with secant lines to $D$ and $E$, I can use the `TanSec_Ang` theorem to find $\angle F$, let $\angle F = V2$:
  `TanSec_Ang V2 V1 N1`
  (This calculates $V2 = (N1 - V1)/2$, using the relationship between the angle formed by secant lines from an external point and the intercepted arcs)
- The final answer is obtained by:
  `Get V2`

4. Formal Language Solution:

```
Sum V0 N0 C180 Chord2_Ang V0 V1 N1 TanSec_Ang V2 V1 N1 Get V2
```

with parameter values:

```
N0=89, N1=137
```

EXAMPLE 2

. . .

YOUR TASK

**Problem:** {{Target problem}}
**Image:** {{Image diagram of the target problem}}
**Solution:** {{Program solution of the target problem}}
**Step-by-Step Thinking:**

### D.3 WEAKNESS OF FORWARD PROMPTING

We prompt the LVLM using manually crafted interleaved formal-natural CoT demonstrations paired with target questions. However, we find that this method often fails. As demonstrated in the following cases, the state-of-the-art LVLMs could revert to natural language instead of maintaining formal language usage during critical reasoning steps, and they often employ operators undefined in our formal language system. These observations suggest that the model inherently lacks knowledge of geometric formal language.

---

**Case 1: Use Natural Language Rather Than Formal Language (Generated OpenAI-o4-mini)**

**Problem:**
$m\angle DCB = 70$, $m\angle BDC = 4x$, $m\angle ABC = 10x + 22$, find the value of $x$ .
**Image:**

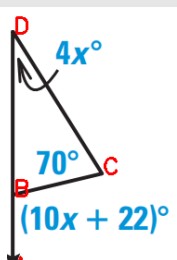

**Output:**
Step-by-step reasoning:
1. Observe that $A$, $B$, and $D$ are collinear with $B$ between $A$ and $D$, so $\angle ABC$ is the exterior angle at $B$ of $\triangle BCD$.
2. In any triangle, an exterior angle equals the sum of the two remote interior angles. Here

$$\angle ABC \ = \ \angle BDC \ + \ \angle DCB.$$

3. Substitute the given measures:

$$10x + 22 \ = \ 4x + 70.$$

4. Solve for $x$:

$$10x + 22 = 4x + 70 \quad \implies \quad 6x = 48 \quad \implies \quad x = 8.$$

Final answer:
$$x = 8.$$

**Error:**
The model directly use natural language to solver the problem without following the instruction prompt.

---

**Case 2: Use Undefined Operators (Generated by Qwen2.5-VL-72B)**

**Problem:**
quadrilateral $DEFG$ is a rectangle. if $m\angle EFD = 2x - 3$ and $m\angle DFG = x + 12$, $GF \perp DG$ on $G$ , find $m\angle EFD$.
**Image:**

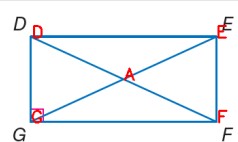

**Output:**
...

Formal Language Solution:

```
Sum (Multiple C2 N0 CMinus C3) (Sum N0 C12) C90 Get V0
```

with parameter values:

```
N0=27
```

**Error:**
`CMinus` is an undefined operator in the formal language.

## D.4 EXAMPLES OF INTERLEAVED FORMAL-NATURAL COT DATASET

We provide some examples in our synthetic interleaved formal-natural CoT dataset below.

### Interleaved Formal-Natural CoT Example1

**Problem:**
assume that all polygons that appear to be regular are regular . $GC = 8$ , find the area of the shaded region .

**Image:**

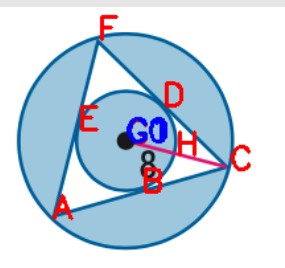

**Output:**
First, label the given length `N0 = 8` (GC is the radius of the outer circle).

1. Compute the area of the outer circle (radius `N0`):
   - `Circle_R_Area N0 V0`
   - $V0 = \pi \cdot \text{N0}^2$

2. Compute the area of the regular triangle inscribed in that outer circle. A regular 3-gon with circumradius `N0` has area:
   - `RNgon_L_Area C3 N0 V1`
   - $V1 = \frac{3 \cdot \text{N0}^2 \cdot \sin(120°)}{2}$

3. Find the apothem (inradius) `V2` of that same triangle by using its area `V1`:
   - `RNgon_H_Area C3 V2 V1`
   - $V1 = 3 \cdot \text{V2}^2 \cdot \tan(60°) \Rightarrow$ `V2` is the apothem

4. Compute the area of the inner circle (incircle of the triangle) of radius `V2`:
   - `Circle_R_Area V2 V3`
   - $V3 = \pi \cdot \text{V2}^2$

5. The shaded region consists of (outer circle minus the triangle) plus the inner circle. First form the difference outer - triangle:
   - `Sum V1 V4 V0`
   - $V1 + V4 = V0 \Rightarrow V4 = V0 - V1$

6. Then add in the inner circle:

- `Sum V3 V4 V5`
  - $V3 + V4 = V5$

7. Finally output the total shaded area `V5`:
   - `Get V5`

Putting it all together:

```
Circle_R_Area N0 V0 RNgon_L_Area C3 N0 V1 RNgon_H_Area C3 V2 V1
    Circle_R_Area V2 V3 Sum V1 V4 V0 Sum V3 V4 V5 Get V5
```

with

```
N0=8
```

---

## Interleaved Formal-Natural CoT Example2

**Problem:**
$QP \parallel RT$, $QR = 2$, $SR = PT = 3$, $PQ = x$, $m\angle TQP = m\angle RQT$, solve for $x$.
**Image:**

**Output:**
Here is the step-by-step setup using our formal language:
Define operands:

- `N0 = QR = 2`
- `N1 = SR = PT = 3`
- `N2 = PQ = x`

1. Since QP ∥ RT, by similar triangles we have ST/PT = SR/QR. Let ST = `V0`:
   - **Operator**: `Proportion V0 N1 N1 N0`
   - This establishes $\frac{V0}{N1} = \frac{N1}{N0}$
2. QS is the sum of QR and RS. Let QS = `V1`:
   - **Operator**: `Sum N0 N1 V1`
   - Calculates $V1 = N0 + N1 = 2 + 3 = 5$
3. In △PQS, QT bisects ∠PQS, so PT/ST = PQ/QS:
   - **Operator**: `Proportion N1 V0 N2 V1`
   - Establishes $\frac{N1}{V0} = \frac{N2}{V1}$
4. Solve for PQ (`N2`):
   - **Operator**: `Get N2`
   - Retrieves the value of `N2` $= x$

Final formal-language solution and parameters:

```
Proportion V0 N1 N1 N0 Sum N0 N1 V1
```

```
Proportion N1 V0 N2 V1 Get N2
```

```
N0=2 N1=3 N2=x
```

# E    ADDITIONAL IMPLEMENTATION DETAILS

## E.1    SUPERVISED FINE-TUNING

**Datasets.** Supervised Fine-Tuning is performed on the formal-integrated CoT dataset synthesized from three geometry problem-solving benchmarks: PGPS9k, UniGeo, and Geo170k. The final dataset comprises 11,000 high-quality synthetic reasoning chains, with contributions of 5,000 from PGPS9k, 2,200 from UniGeo, and 3,800 from Geo170k.

**Training Details.** We fine-tune the Qwen-2.5-VL-7B-Instruct model implemented via LLaMA-Factory framework[7], running for 2 epochs on 4 NVIDIA A100 80GB GPUs. The key training parameters are as follows:

```
torchrun --nproc_per_node 4 src/train.py \
    --finetuning_type full \
    --do_train \
    --adam_beta2 0.95 \
    --model_name_or_path Qwen2.5-VL-7B-Instruct \
    --trust_remote_code \
    --preprocessing_num_workers 8 \
    --template qwen2_vl \
    --warmup_ratio 0.03 \
    --weight_decay 0.0 \
    --per_device_train_batch_size 8 \
    --gradient_accumulation_steps 4 \
    --gradient_checkpointing \
    --ddp_timeout 9000 \
    --learning_rate 2e-5 \
    --lr_scheduler_type cosine \
    --logging_steps 2 \
    --cutoff_len 4096 \
    --num_train_epochs 2 \
    --bf16 \
    --seed 42 \
    --flash_attn fa2 \
```

**Compute Resources.** The SFT stage runs using 4 NVIDIA A800-80GB GPUs. The training process completes in approximately 0.5 hours.

## E.2    REINFORCEMENT LEARNING

**Datasets.** Reinforcement Learning is performed on 19.5k samples drawn from the training splits of three geometry problem-solving benchmarks: PGPS9k, UniGeo, and Geo170k. The distribution consists of 8k samples from PGPS9k, 3.5k from UniGeo, and 8k from Geo170k.

**Training Details.** This stage builds upon the model initialized from supervised fine-tuning. We employ the GRPO algorithm (Shao et al., 2024) implemented in the verl framework[8]. We run for 15 epochs on 8 NVIDIA H20 96GB GPUs. Key hyperparameters configured as follows:

```
python3 -m verl.trainer.main_ppo \
    algorithm.adv_estimator=grpo \
    data.train_batch_size=256 \
    data.max_prompt_length=2048 \
    data.max_response_length=1024 \
    data.val_batch_size=5120 \
    data.filter_overlong_prompts=True \
```

---

[7]https://github.com/hiyouga/LLaMA-Factory
[8]https://github.com/volcengine/verl

```
      data.truncation='error' \
      data.image_key=images \
      actor_rollout_ref.model.path=$MODEL_PATH \
      actor_rollout_ref.actor.optim.lr=1e-6 \
      actor_rollout_ref.actor.optim.lr_warmup_steps=0 \
      actor_rollout_ref.actor.optim.weight_decay=0.0 \
      actor_rollout_ref.model.use_remove_padding=True \
      actor_rollout_ref.actor.ppo_mini_batch_size=64 \
      actor_rollout_ref.actor.ppo_micro_batch_size_per_gpu=16 \
      actor_rollout_ref.actor.use_kl_loss=True \
      actor_rollout_ref.actor.kl_loss_coef=0.0 \
      actor_rollout_ref.actor.kl_loss_type=low_var_kl \
      actor_rollout_ref.actor.entropy_coeff=0 \
      actor_rollout_ref.actor.clip_ratio_high=0.28 \
      actor_rollout_ref.model.enable_gradient_checkpointing=True \
      actor_rollout_ref.actor.fsdp_config.param_offload=False \
      actor_rollout_ref.actor.fsdp_config.optimizer_offload=False \
      actor_rollout_ref.rollout.log_prob_micro_batch_size_per_gpu=64 \
      actor_rollout_ref.rollout.tensor_model_parallel_size=1 \
      actor_rollout_ref.rollout.name=vllm \
      actor_rollout_ref.rollout.gpu_memory_utilization=0.8 \
      actor_rollout_ref.rollout.enable_chunked_prefill=False \
      actor_rollout_ref.rollout.enforce_eager=False \
      actor_rollout_ref.rollout.free_cache_engine=False \
      +actor_rollout_ref.rollout.enable_prefix_caching=True \
      actor_rollout_ref.rollout.n=$n_samples_per_prompt \
      actor_rollout_ref.rollout.temperature=1.0 \
      actor_rollout_ref.rollout.val_kwargs.temperature=0.0 \
      actor_rollout_ref.rollout.val_kwargs.n=1 \
      actor_rollout_ref.rollout.val_kwargs.do_sample=False \
      actor_rollout_ref.ref.log_prob_micro_batch_size_per_gpu=64 \
      actor_rollout_ref.ref.fsdp_config.param_offload=True \
      algorithm.kl_ctrl.kl_coef=0.0 \
      custom_reward_function.path=./verl/utils/reward_score/pgps9k.py \
      custom_reward_function.name=compute_score \
      trainer.val_before_train=True \
      trainer.critic_warmup=0 \
      trainer.n_gpus_per_node=8 \
      trainer.nnodes=1 \
      trainer.total_epochs=15 \
```

**Compute Resources.** The RL training runs using 8 NVIDIA H20-96GB GPUs. The training process completes in approximately 83 hours.

## F  LIMITATIONS

While our geometric formal language paradigm demonstrates significant improvements in reducing computational and reasoning errors, it is not perfect with limitations. First, the current framework cannot handle problems requiring diagrammatic constructions (e.g., adding auxiliary lines), which restricts its applicability to certain classes of geometry problems. Second, our method cannot perform long-horizon reasoning with self-reflection as in (Huang et al., 2025) yet. We believe empowering our framework with this feature would further enhance the performance. Additionally, our approach receives limited reward supervision during training, in contrast to the rich executor feedback and multi-turn interaction mechanisms used by (Li et al., 2025a). Exploring these enhancements is a promising direction for future work.

