# OpenReview forum: "Bridging Formal Language with Chain-of-Thought Reasoning to Geometry Problem Solving"
_ICLR.cc/2026/Conference — ICLR 2026 Conference Desk Rejected Submission_

### Official Review · Reviewer_8pkS · 2025-10-28

**Soundness:** 3
**Presentation:** 3
**Contribution:** 2
**Rating:** 4
**Confidence:** 4

**Summary:**

To address the unreliable computation of LVLMs in Geometry Problem Solving (GPS), this paper proposes a "hybrid reasoning" framework. This framework combines natural language Chain-of-Thought (CoT) with executable formal code, trained in two stages (SFT+RL) on a new 11K-sample dataset. The resulting GF-Reasoner, based on Qwen2.5-VL-7B, outperforms the larger Qwen2.5-VL-72B and achieves more reliable and concise solution.

**Strengths:**

1.	The method achieves excellent performance and efficiency. The GF-Reasoner (7B) significantly outperforms the 10x larger Qwen2.5-VL-72B by +15.2% (PGPS9K) and +4.8% (UniGEO). This hybrid reasoning approach also reduces the "computation error" rate to nearly zero (0.3%) and substantially decreases the "reasoning error" rate from 23.0% to 14.3%.
2.	The paper validates an effective "SFT + solver-in-the-loop RL" framework, with the RL stage alone improving Pass@1 accuracy by +26.6%. The detailed ablation studies offer valuable insights by analyzing the impact of data synthesis strategies, SFT training epochs, and CoT style.

**Weaknesses:**

1.	The core method (CoT + external solver) is a well-established paradigm (e.g., AlphaGeometry, Toolformer, PoT, Logic-RL, ReTool, ToRL, Search-O1). The reliance on an existing solver (PGPSNet's 34 operations) shifts the contribution to data synthesis and an SFT+RL pipeline, rather than methodological innovation.
2.	The framework is highly specialized, confined to a closed set of 34 predefined solver operations. This limits scalability (e.g., cannot "add auxiliary lines" per Section F) and lacks essential general tools (image processing, Python, web search, API). The absence of image tools might explain the higher visual perception error (8.0% vs 3.0%) reported in Table 2.
3.	The comparisons are insufficient, omitting recent models (e.g., Qwen3-VL, GLM-4.5V, Gemini 2.5 Pro, GPT5) and relevant visual-driven methods (e.g., "Thinking with Images").

**Questions:**

1.	Given that the "CoT + external solver" and "SFT + RL" paradigms are well-established, what is the core methodological innovation of this paper?
2.	Table 2 shows your model's visual perception error is higher (8.0% vs 3.0%). Could this be due to lacking image processing tools (e.g., zoom/crop), and have you considered integrating them?
3.	What are the advantages of this method over recent visual planning approaches like "Thinking with Images," and can you provide comparisons to justify the complex training pipeline?

---

> ### Author Response · Authors · 2025-11-24
> **Response (Part 1)**
>
> **Comment 1:** The core method (CoT + external solver) is a well-established paradigm (e.g., AlphaGeometry, Toolformer, PoT, Logic-RL, ReTool, ToRL, Search-O1). The reliance on an existing solver (PGPSNet's 34 operations) shifts the contribution to data synthesis and an SFT+RL pipeline, rather than methodological innovation.
>
> **Response 1:**
>
> We thank the reviewer for this insightful comment. We agree that a significant aspect of our contribution lies in the novel data synthesis strategy and the SFT+RL pipeline, which we believe are important for advancing this field. We would like to clarify our methodological innovation beyond this:
>
> 1.  We introduce an **interleaved formal-natural reasoning paradigm**, where natural language reasoning and solver-executable operators alternate in synchronized steps within a single chain of thought. This differs from existing works in geometry problem solving:
>    - Pure natural language reasoning (standard CoT, e.g., Qwen2-VL series, Claude, GPT)
>    - Pure formal programs (traditional symbolic solvers like GeoX and AlphaGeometry)
>
> The paradigm combines the flexibility of natural language and preciseness of formal language reasoning. We also provide in-depth analysis of how interleaving maintains exploration potential and reduces reasoning and computation errors.
>
> 2. While works like ReTool and ToRL focus on leveraging models' existing Python/tool-use capabilities (which involve interleaving natural language with natural language actions like API calls), our work aims to teach LVLMs the novel capability of domain-specific symbolic reasoning. This requires them to interleave natural language with formal symbolic operations, a process that demands far stricter precision.
>
> **Comment 2:** The framework is highly specialized, confined to a closed set of 34 predefined solver operations. This limits scalability (e.g., cannot "add auxiliary lines" per Section F) and lacks essential general tools (image processing, Python, web search, API). The absence of image tools might explain the higher visual perception error (8.0% vs 3.0%) reported in Table 2.
>
> **Response 2:** Thank you for this comment.
>
> - We note that "adding auxiliary lines," "image processing," and "geometric operators" represent **different axes** for addressing geometry problem-solving. While we value these capabilities, we believe they are orthogonal contributions, and future work could combine them to further reduce errors.
>
> -  In this paper, we focus on 34 domain-specific mathematical operators designed for geometry reasoning. This choice, also adopted by works like GeoX, **enables a controlled study** that isolates the effectiveness of our interleaved formal-natural reasoning paradigm and ensures fair comparison.
>
> - On visual perception errors: We acknowledge the higher visual error rate (8.0% vs. 3.0%) and agree that image-processing tools present a promising complementary direction. To this end, we conducted a preliminary experiment by adding a zoom function. As detailed in Appendix C (Zooming Images for Better Perception), this single intervention reduced the visual perception error from 8.0% to 6.0% and increased the overall accuracy on PGPS9K from 68.7% to 69.3%.

---

> ### Author Response · Authors · 2025-11-24
> **Response (Part 2)**
>
> **Comment 3:** The comparisons are insufficient, omitting recent models (e.g., Qwen3-VL, GLM-4.5V, Gemini 2.5 Pro, GPT5) and relevant visual-driven methods (e.g., "Thinking with Images").
>
> **Response 3:** Thank you for this valuable suggestion. We have evaluated these recent models on PGPS9k dataset and provide updated comparisons below:
>
> | Model                              | Accuracy | Release Time |
> | ---------------------------------- | -------- | ------------ |
> | Qwen2.5-VL-7B-Instruct             | 39.4%    | 28/01/2025   |
> | Qwen2.5-VL-32B-Instruct            | 50.0%    | 25/03/2025   |
> | Qwen3-VL-8B-Instruct               | 70.9%    | 15/10/2025   |
> | GLM-4.5V (106B)                    | 77.9%    | 18/08/2025   |
> | Gemini 2.5 Pro                     | 83.3%    | 17/06/2025   |
> | GPT-5                              | 85.8%    | 08/08/2025   |
> | "Thinking with Images" (OpenAI-o3) | 77.2%   | 16/04/2025   |
> | **Ours (Qwen2.5-VL-7B-Instruct)**                         | 68.7%    |     -      |
> | **Ours (Qwen2.5-VL-32B-Instruct)**                        | 82.1%    |     -         |
>
> - **Observations:**
>      - Our 32B model achieves competitive performance with state-of-the-art proprietary models (Gemini 2.5 Pro, GPT-5, OpenAI-o3), despite being significantly smaller. While GPT-5 achieves higher accuracy, we note that direct comparison is challenging due to unknown model sizes and training data for these proprietary systems.
>      - Importantly, our results demonstrate clear scaling effects—performance improves substantially from 7B to 32B—suggesting that our interleaved reasoning framework could benefit further from larger base models.
>
> * **Note on timing:** Several of these models (Qwen3-VL, GLM-4.5V, GPT-5) were released within one month of or after the ICLR submission deadline, which is why they were not included in our original comparison. We appreciate the opportunity to provide these updated results.
>
> ****
> **Question 1:** Given that the "CoT + external solver" and "SFT + RL" paradigms are well-established, what is the core methodological innovation of this paper?
>
> **Response 1:** Thanks for raising this discussion. Our work makes several contributions beyond "CoT + external solver" and "SFT + RL" paradigms:
>
> **1. New reasoning paradigm**: We propose a new hybrid reasoning paradigm for GPS that combines the strengths of formal and natural language reasoning. We also provide in-depth analysis of how interleaving maintains exploration potential and reduces reasoning and computation errors.
>
> **2. Curated training data**: We propose a novel customized backward data synthesis strategy that improves interleaved CoT synthesis accuracy by 30.5% over forward synthesis, contributing an interleaved formal-natural CoT dataset specifically designed for geometry, filling an important gap in the community.
>
> **3. Systematic training strategies**: We conduct a comprehensive study of SFT and RL approaches for realizing interleaved formal-natural reasoning paradigm, revealing optimal training strategies and the balance between supervised fine-tuning and reinforcement learning.
>  ***
> **Question 2:** Table 2 shows your model's visual perception error is higher (8.0% vs 3.0%). Could this be due to lacking image processing tools (e.g., zoom/crop), and have you considered integrating them?
>
> **Response 2:** Thanks for presenting a promising complementary direction. To this end, we conducted a preliminary experiment by integrating a zoom tool and found it effective. It provides higher-resolution details, which improves visual perception. On PGPS9K, this raised accuracy from 68.7% to 69.3% and reduced the visual perception error from 8.0% to 6.0%. A supporting case study is provided in Figure 11, where applying a 2x zoom enables the model to obtain a clearer view, leading to the correct identification of corresponding sides in a pair of similar triangles. These results suggest that incorporating image processing tools can aid visual reasoning and improve geometric understanding, thereby complementing the current framework.

---

> > ### Author Response · Authors · 2025-11-24
> > **Response (Part 3)**
> >
> > **Question 3:** What are the advantages of this method over recent visual planning approaches like "Thinking with Images," and can you provide comparisons to justify the complex training pipeline?
> >
> > **Response 3:** We thank the reviewer for this insightful question. We clarify that our framework is orthogonal and complementary to "Thinking with Images":
> >
> > 1. **"Thinking with Images"**: Emphasizes visual representations and planning in the *image space*, leveraging visual reasoning for geometric understanding.
> >
> > - **Our approach**: Proposes a new reasoning paradigm in the *language space* that interleaves formal symbolic computation with natural language reasoning, aiming to achieve both mathematical precision (through formal language) and reasoning flexibility (through natural language).
> >
> > Therefore, these two frameworks are *not conflicting but orthogonal*  – they focus on different representation spaces.  An exciting future direction would be to combine both paradigms. We will explore this integration in future work. We have added this discussion in Appendix B.
> >
> > ***
> > Thank you for your thoughtful and constructive feedback. We believe our revisions have addressed your concerns and hope you might consider updating your review score accordingly. We are happy to provide any additional clarification if needed.

---

> > > ### Comment · Reviewer_8pkS · 2025-11-26
> > >
> > > I have read the authors' rebuttal and will maintain my scores.

---

> > > > ### Author Response · Authors · 2025-11-27
> > > > **Response to Reviewer 8pkS's Comment**
> > > >
> > > > We thank the reviewer for reading our rebuttal.
> > > >
> > > > However, we respectfully note that our rebuttal provided **2 new experimental results** and **3 related work discussions** specifically aimed at your original concerns regarding method effectiveness and comparisons with prior work. We believe these results directly resolve the issues raised in the initial review.
> > > >
> > > > Given that these additions were made directly in response to your initial comments, we are unsure why you decide to maintain the review score.  **Could you please provide specific feedback on why these new results are insufficient to change your assessment?** This would greatly help us understand the gap between our revision and your expectations. This is also important for us to improve the paper quality and for the community to advance the field. We are happy to provide further clarification on remaining concerns.

---

### Official Review · Reviewer_2ABk · 2025-10-30

**Soundness:** 3
**Presentation:** 3
**Contribution:** 3
**Rating:** 4
**Confidence:** 3

**Summary:**

This paper presents a hybrid reasoning approach to solve geometry problems for LVLMs, via combining natural and formal languages. They also curate a dataset of 11K questions and used it in post-training SFT and RL for training purposes. Their model beats the SOTA on relevant benchmarks.

**Strengths:**

1. This is a nice and well-written paper that tackles an interesting problem that LVLMs face.

2. The hybrid reasoning approach presented in the paper is novel where it combines interpretability and precision of the two core components, i.e., natural language CoT and formal reasoning.

3. Training with SFT and RL shows robust framework and performance boost. Their 7B model outperforms the peers as well as larger model of 72B.

4. The paper contributed to the field by curating a synthetic dataset of 11K questions.

5. Authors conducted extensive experiments including ablation studies.

**Weaknesses:**

1. One major concern that I have is that there are only 34 operators which means if a problem needs a geometry theorem, it is not included in the set.

2. Dataset was essentially distilled from large and strong teacher models, hence model performance is heavily dependent on these models.

3. RM is quite simple (binary) and the model doesn't take partial credit for correct steps rather than final answer.

**Questions:**

1. OOD theorems: what happens if the model encounters a problem that isn't part of the 34 operators? Is the model able to learn how to derive theorems that are not included in the set?

2.  I wonder what happens if the RM generates continuous reward rather than a simple binary where the model can be rewarded partially for a correct step?

3. Is there a mechanism to check if the performance boost is due to the model being distilled from powerful teacher models or it's a natural result of an RL step?

---

> ### Author Response · Authors · 2025-11-24
> **Response (Part 1)**
>
> We sincerely appreciate your kind words and thoughtful assessment.
>
> **Comment 1 :** One major concern that I have is that there are only 34 operators which means if a problem needs a geometry theorem, it is not included in the set.
>
> **Response 1:**  Thank you for raising this important concern. We acknowledge that our work builds upon the 34 operators introduced in Zhang et al. (2023a).  It has been used extensively in prior systems such as GEOX, PGPSNet, and LANS. Empirically, these operators already demonstrate strong effectiveness in solving the geometry problems in existing benchmarks.
>
> We agree that expanding the operator set to include additional geometric theorems is a valuable direction. However, such expansion is **orthogonal to the focus of our paper**.
>
> Our goal is to introduce a **new** **reasoning** ***paradigm*** for geometry problem solving,  aiming to achieve both mathematical precision (through formal language) and reasoning flexibility (through natural language).
>
> Notably, the proposed framework is designed to be scalable and can readily accommodate a broader set of formal operators. This flexibility makes it well-suited for future work aimed at extending the formal language.
>
> More broadly, if the long-term objective is to train models that can use symbolic operators with human-like flexibility, a prerequisite is to develop methods that ensure the model can ***reliably leverage the available operators***. We believe our work achieves this initial step: it provides an effective way for LVLMs to internalize the semantics of existing operators, plan their usage through CoT, and integrate them into solver-executable reasoning traces. This capability, in turn, may inform future efforts on designing richer or more expressive operator sets.
>
>
>
> **Comment 2:** Dataset was essentially distilled from large and strong teacher models, hence model performance is heavily dependent on these models.
>
> **Response 2:**  Thank you for raising this discussion. We would like to clarify that our approach fundamentally differs from standard knowledge distillation in several key aspects:
>
> - Our data synthesis process incorporates external validation signals beyond teacher outputs. Teacher-generated reasoning chains are executed through an independent geometry solver to verify correctness. Only chains producing correct answers are retained for training. This external validation fundamentally shapes the data distribution generated by the teacher model.
> - The teacher model is used only during the initial SFT phase for cold-start initialization. The subsequent RL training phase is completely teacher-independent, relying on the feedback from an external solver.
>
> These differences make our training process not simple distillation.  Using interleaved CoT to solve geometry problems, our trained model demonstrates superior performance (68.8%) compared to teacher (OpenAI-o4-mini) model's performance of 37.6%.
>
> **Comment 3:** RM is quite simple (binary) and the model doesn't take partial credit for correct steps rather than final answer.
>
> **Response 3:**  Following your advice, we investigated extending the binary final-answer reward with partial credit for intermediate steps.
>
> In our preliminary studies (not shown in the submission), we have explored fine-grained rewards  that checks the *grammatical validity* of the generated formal program (i.e., reward=1 if the program is syntactically correct, else 0). The performance during RL training is shown in Figure 10 in Appendix C [On the Design of Reward Function].   Final accuracy is shown below:
>
> | Reward design    | PGPS9K | UniGEO |
> | ---------------- | ------ | ------ |
> | Binary           | **68.7%**  | **72.7%** |
> | Binary + Process | 66.6%  | 70.2%  |
>
> The results indicate that adding this process reward during RL training dose not yield gains compared to using the binary outcome reward alone. This indicates that designed partial credit may not be necessary, which may be due to two reasons:
>
> - As the RL training uses multiple trajectory sampling per problem, correct intermediate steps appear more frequently in successful trajectories. This provides **implicit partial credit** through statistical aggregation., enabling the model to learn which intermediate steps correlate with correct outcomes. As also noted by Wen et al. [1], RL with verifiable rewards implicitly incentivizes valid reasoning even when rewards are based solely on final-answer correctness.
> - While process-level rewards (partial credit for intermediate steps) seem appealing, they may also introduce reward-hacking challenges: models may learn to generate superficially correct-looking steps without genuine reasoning, a phenomenon also discussed in DeepSeek-R1 [2].
>
> [1] Wen et al. Reinforcement Learning with Verifiable Rewards Implicitly Incentivizes Correct Reasoning in Base LLMs, NeurIPS 2025.
> [2] DeepSeek-R1: Incentivizing Reasoning Capability in LLMs via Reinforcement Learning, Nature 2025.

---

> ### Author Response · Authors · 2025-11-24
> **Response (Part 2)**
>
> **Question 1:** OOD theorems: what happens if the model encounters a problem that isn't part of the 34 operators? Is the model able to learn how to derive theorems that are not included in the set?
>
> **Response 1:**  Thank you for this insightful question. Our work focuses on training a specialized geometry reasoning model using a predefined set of 34 geometric operators, similar to the approach taken by other recent geometry solvers. By design, the model's reasoning is constrained to this operator set, meaning it cannot solve problems requiring theorems outside this predefined collection.
>
> This limitation could be addressed in future work through two approaches: (1) expanding the operator set to include additional theorems, or (2) training models to dynamically generate new operators when encountering novel problem types—potentially achievable through RL with appropriate reward incentives.
>
> In this paper, we focus on the foundational challenge of designing effective post-training methods that enable models to reliably leverage existing operators for geometric reasoning, which could inspire future work in this direction.
>
>
> **Question 2:** I wonder what happens if the RM generates continuous reward rather than a simple binary where the model can be rewarded partially for a correct step?
>
> **Response 2:** Thanks for the advice. We have responded to this matter in Response 2 of Part I.
>
>
> **Question 3:** Is there a mechanism to check if the performance boost is due to the model being distilled from powerful teacher models or it's a natural result of an RL step?
>
> **Response 3:** Thanks for raising this question. To disentangle contributions from teacher distillation vs. RL training, we evaluate: (1) **SFT** **model**: Fine-tuned on filtered teacher trajectories. (2) **SFT + RL model**: Further trained with solver-in-the-loop RL .
>
> |          | PGPS9K | UniGEO |
> | -------- | ------ | ------ |
> | SFT      | 42.1%  | 46.0%  |
> | SFT + RL | **68.7%**  | **72.7%**  |
>
> According to the results, RL provides substantially higher performance than the initial SFT, which enables the the model to discover more effective reasoning patterns through exploration.
>
> ****
> We sincerely thank you for your thoughtful and constructive feedback. We hope the clarifications and additional analyses in our revised manuscript have addressed your concerns. If so, we would be grateful if you could consider updating your review score accordingly. Please do not hesitate to let us know if you have any further questions.

---

> > ### Comment · Reviewer_2ABk · 2025-11-25
> >
> > Thanks for addressing my concerns and clarifications. I am happy to increase my score.

---

> > > ### Author Response · Authors · 2025-11-27
> > > **Response to Reviewer 2ABk's Comment**
> > >
> > > Thank you very much for your feedback. We're glad our responses have resolved your concerns!

---

### Official Review · Reviewer_VSGS · 2025-10-30

**Soundness:** 2
**Presentation:** 3
**Contribution:** 2
**Rating:** 4
**Confidence:** 2

**Summary:**

The paper introduces GF-Reasoner, a fine-tuned large vision-language model (LVLM) built on Qwen2.5-VL 7B that combines formal language programs with natural language chain-of-thought (CoT) for geometry problem solving. The model alternates between descriptive reasoning and executable formal code. The authors construct a new 11k-sample synthetic dataset with interleaved formal-natural reasoning and employ supervised fine-tuning (SFT) on the synthetic dataset and solver-integrated reinforcement learning (RL), where correctness feedback comes from a geometric solver. GF-Reasoner achieves up to 15% higher accuracy compared to both specialist geometry solvers and much larger LVLMs.

**Strengths:**

- **Clear motivation:** The paper addresses geometry problem solving, which is difficult for the current LVLMS.
- **Rigorous ablations:** The study isolates the effects of the added elements such as the interleaved CoT, synthesis strategies, and RL. This helps the readers understand the contribution of each elements to the performance gain.

**Weaknesses:**

- **Questionable necessity of the proposed approach**
    - The paper does not demonstrate that the hybrid approach is fundamentally better than just improving pure natural language reasoning (SFT + RL only on natural language reasoning).
    - To add, Interleaving CoT is not a novel idea. This is similar to how the ReAct framework works.
- **The performance comparisons are not particularly fair**
    - The proposed model has access to formal language specs, while the other baseline models do not have access to those. Thus, it is difficult to completely isolate the factors that lead to the improvement. The simpler conclusion may be that the comparisons measure performance given access to tools, instead of the “bridging of formal language and CoT”.
- **The analyses lack explanations**
    - How did the authors classify the error types presented in Table 2 and what are their definitions? Is this an exhaustive list of errors? Who annotated these errors and are there inter-annotator agreement statistics?
    - Did the authors evaluate the checkpoints after the SFT epochs on the benchmarks beyond the PGPS9K? It is interesting to note that even in the PGPS9K, the 2 epochs SFT initialization achieves higher Accuracy than the others (Figure 6a, left-most points), potentially suggesting overfitting.

**Questions:**

See Weaknesses.

Note that I am not an expert in this research area. I carefully checked the paper for the details; however, there may be prior studies that I am not familiar with which may provide more context to the general contributions of this paper.

---

> ### Author Response · Authors · 2025-11-24
> **Response (Part 1)**
>
> Thank you for your thoughtful and positive review of our work. We appreciate your encouraging feedback and have addressed your questions below.
>
> **Commen1.1**: Questionable necessity of the proposed approach. The paper does not demonstrate that the hybrid approach is fundamentally better than just improving pure natural language reasoning (SFT + RL only on natural language reasoning).
>
> **Response 1.1:** Thank you for this important question about the necessity of our approach. We would like to clarify that our paper already includes comprehensive comparisons demonstrating the advantages of interleaved CoT over natural language reasoning. Specifically, in the second paragraph of Section 4.3 (Section 3.3 in the submission version), we compare SFT+RL models using interleaved CoT versus pure natural language CoT. As shown in Table 3, interleaved CoT demonstrates reductions in reasoning errors, geometry knowledge errors, and computation errors compared to pure natural language alone. Error analysis in Appendix C further illustrates how the interleaved approach specifically reduces computational and reasoning errors.
>
> To provide a more direct comparison, in Appendix C's Reasoning Paradigm Comparison, we compare interleaved CoT with pure natural language reasoning under identical training conditions . It achieves +8.7% on PGPS9k and +4% on UniGEO, confirming its clear advantage.
>
>
> **Comment 1.2:** To add, Interleaving CoT is not a novel idea. This is similar to how the ReAct framework works.
>
> **Response 1.2:**  Thank you for this observation. We acknowledge the high-level conceptual similarity to ReAct in terms of interleaving reasoning and actions. However, there are fundamental differences in problem setting, technical challenges, and contributions.
>
> - **Inference-time prompting vs. Learning interleaved reasoning capabilities:**
>
> ReAct is primarily an inference-time prompting framework that enables pre-trained models to alternate between reasoning and acting for general task solving. It assumes the model already possesses the necessary reasoning and action capabilities, and focuses on how to better orchestrate them through prompting. However, as shown in the table below (added to Appendix C, Table 7), directly prompting state-of-the-art models to perform interleaved formal-natural reasoning in geometry yields poor results:
>
> | Model                    | Qwen-2.5-7B-VL | Qwen-2.5-72b-VL | Claude-3.7-sonnet | GPT-4o | **Ours (Post-trained)** |
> | ------------------------ | -------------- | --------------- | ----------------- | ------ |------ |
> | Acc. of Direct Prompting | 0.1%           | 14.03%          | 16.2%             | 6.4%   | **68.7%**|
>
> In contrast, our work addresses a fundamentally different challenge: **teaching models to acquire the capability** to perform interleaved formal-natural reasoning through post-training. Geometry problem solving requires models to learn domain-specific formal operators and how to properly interleave them with natural language reasoning—capabilities that do not exist in base models.
>
> - **Natural language actions vs. Formal symbolic operations:**
>
> ReAct interleaves natural language reasoning with natural language actions (e.g., "Search[topic]" or API calls). Our approach interleaves natural language with **formal symbolic operations** that require: strict syntactic correctness; symbolic precision and mathematical rigor; domain-specific geometric operators (e.g., angle calculations, geometric constructions).  This presents distinct technical challenges in data curation, training stability, and ensuring both reasoning coherence and formal correctness.
>
> - **Our contributions:**
>
> Our work makes several contributions beyond applying an interleaving concept: 1) **Curated** **training data**: A interleaved formal-natural  CoT dataset specifically designed for geometry, filling a gap in the community;  2) **Systematic training strategies**: Comprehensive study of SFT and RL approaches for learning interleaved reasoning;  3) **Domain-specific** **insights**: Analysis of how interleaving maintains exploration potential and reduces reasoning and computation errors in mathematical problem solving.
>
> That said, while ReAct demonstrates the value of interleaving at inference time, our work shows how to effectively train models to perform interleaved formal-natural reasoning in specialized mathematical domains. These are complementary contributions addressing different aspects of the broader challenge. We have updated Appendix B in the manuscript to make these points clear.

---

> > ### Author Response · Authors · 2025-11-24
> > **Response (Part 2)**
> >
> > **Comment 2:** The performance comparisons are not particularly fair
> >
> > - The proposed model has access to formal language specs, while the other baseline models do not have access to those.
> > - Thus, it is difficult to completely isolate the factors that lead to the improvement.
> > - The simpler conclusion may be that the comparisons measure performance given access to tools, instead of the “bridging of formal language and CoT”.
> >
> > **Response 2.**  Thanks for raising this discussion.
> >
> > - We would like to clarify that our performance gains stem not merely from tool access. To isolate the this factor, we conducted a controlled experiment:
> >   - We provided baseline models (GPT-4o, Claude-3.7-Sonnet, Qwen-2.5-VL series) with **identical formal language specifications** (used in our training data synthesis) via detailed prompting. This ensures they also have "tool access". As shown below, despite having explicit access to formal specifications, these strong foundation models still struggle to generate valid interleaved CoT.
> >
> > | Model                    | Qwen-2.5-7B-VL | Qwen-2.5-72b-VL | Claude-3.7-sonnet | GPT-4o | **Ours (Post-trained)** |
> > | ------------------------ | -------------- | --------------- | ----------------- | ------ |------ |
> > | Acc. of Direct Prompting | 0.1%           | 14.03%          | 16.2%             | 6.4%   | **68.7%**|
> >
> > - This demonstrates that **tool access alone is insufficient**. The critical factor is teaching models through tailored data synthesis, and post-training to internalize formal reasoning patterns and seamlessly integrate them with natural language CoT.
> >
> > We have updated the results in Appendix C's Performance Gains Beyond Tool Access for clarification.
> >
> >
> > **Comment 3.1**: The analyses lack explanations. How did the authors classify the error types presented in Table 2 and what are their definitions? Is this an exhaustive list of errors? Who annotated these errors and are there inter-annotator agreement statistics?
> >
> > **Response 3.1:** Thanks for the questions.
> >
> > **How did the authors classify the error types presented in Table 2 and what are their definitions? Is this an exhaustive list of errors?:**  Our error taxonomy is adapted from established frameworks in geometry problem-solving research [1, 2], which represent widely adopted categorizations in this domain. The definitions of these four types of errors are:
> >
> > - Visual perception errors: The model failed to either extract salient features from the image or associate the textual information with its corresponding geometric shapes.
> > - Reasoning errors: The model failed to generate a valid solution, demonstrating either a lack of logical coherence and internal consistency or a failure in procedural execution by skipping necessary operations.
> > - Geometric knowledge errors: The model's error was either in misapplying a geometric concept or in failing to distinguish between the definitions of two theorems.
> > - Computation errors: The model failed to calculate the numerical results, leading to errors in calculations involving arithmetic, square roots, and proportions.
> >
> > llustrative examples have been provided in Appendix C [Error Analysis]. As this taxonomy is widely adopted to cover the major error modes observed in geometry problem-solving systems. In our case study of randomly sampled incorrect predictions in Section 4.3, these four categories could cover all errors.
> >
> > [1] Zhang et al. MATHVERSE: Does Your Multi-modal LLM Truly See the Diagrams in Visual Math Problems? ECCV2024
> >
> > [2] Fu et al. GeoLaux: A Benchmark for Evaluating MLLMs’ Geometry Performance on Long-Step Problems Requiring Auxiliary Lines. 2025
> >
> > **Who annotated these errors and are there inter-annotator agreement statistics?**  For annotation, two PhD students and one master student in Computer Science are involved, all with expertise in geometry problem solving and LVLMs. Each annotator independently labeled 300 test cases. Disagreements were resolved through discussion to reach consensus. The initial annotation achieved 91% inter-annotator agreement.
> >
> > Following your suggestion, we have added the definitions of each error type, annotation protocol and agreement statistics to Appendix C [Error Analysis].

---

> > > ### Author Response · Authors · 2025-11-24
> > > **Response (Part 3)**
> > >
> > > **Comment 3.2:** Did the authors evaluate the checkpoints after the SFT epochs on the benchmarks beyond the PGPS9K? It is interesting to note that even in the PGPS9K, the 2 epochs SFT initialization achieves higher Accuracy than the others (Figure 6a, left-most points), potentially suggesting overfitting.
> > >
> > > **Response 3.2:**  Thank you for this insightful observation. We have evaluated checkpoints from different SFT epochs across multiple benchmarks (UniGEO, MathVista, and MathVerse), as shown below:
> > >
> > > | Benchmark | 2 Epochs | 4 Epochs | 6 Epochs | 8 Epochs |
> > > | --------- | -------- | -------- | -------- | -------- |
> > > | UniGEO    | **45.8** | 42.7     | 43.0     | 45.6     |
> > > | MathVista | **43.8** | 40.4     | 39.4     | 36.6     |
> > > | MathVerse | 42.7     | **44.1** | 39.5     | 41.7     |
> > >
> > > The results reveal the following patterns:
> > > - MathVista exhibits clear overfitting, with performance degrading monotonically as SFT epochs increase.
> > > - UniGEO and MathVerse show non-monotonic trends, indicating that more SFT epochs do not consistently yield better performance.
> > >
> > > Importantly, despite these mixed effects on post-SFT performance, we observe that models trained with **more SFT epochs consistently achieve worse final performance after RL training across all three benchmarks (see Figure 8 in Appendix C )**. We attribute this primarily to entropy collapse during extended SFT training, which impairs exploration during the subsequent RL stage and limits the model's exploration space in RL.
> > >
> > >
> > > We sincerely thank you for your thoughtful and constructive feedback. We hope the clarifications and additional analyses in our revised manuscript have addressed your concerns. If so, we would be grateful if you could consider updating your review score accordingly. Please do not hesitate to let us know if you have any further questions.

---

> > > > ### Comment · Reviewer_VSGS · 2025-11-27
> > > >
> > > > I thank the authors for the responses and the additional results provided. However, I would like to still maintain my score. Here are my rationales:
> > > >
> > > > ## Interleaving CoT is not a novel idea, neither training the model to do this interleaving. But beyond the novelty issue, the motivation remains unclear.
> > > >
> > > > I sincerely thank the authors for the explanation. However, I think the authors might have dismissed the fact that fine-tuning a model to do this interleaving process is also not necessarily novel. There is a section in the ReAct paper that discusses fine-tuning. Not to mention, agentic fine-tuning that involves multi-turn/interleaving is becoming more mainstream.
> > > >
> > > > However, I do not want to dwell on the question of novelty. Because even if a topic is not novel, a paper can still be a solid body of work. Unfortunately, I think there are still missing pieces in this work that may limit its contribution. Particularly because some of the design decisions may look arbitrary and there may be a gap between the experimental results and the major claims, including:
> > > > - Why does it have to be a natural language and a specific formal language that are interleaved? Can it be any arbitrary formal language?
> > > > - The claim that this training process helps models to "internalize formal reasoning patterns and seamlessly integrate them with natural language CoT" sounds like an overclaim. Another simplest explanation would be that we are teaching this new exotic language to the model, and perhaps this specific geometry solver language is suitable for the problems at hand, but may not be suitable for other problems. "Internalizing formal reasoning patterns" sounds like a broader claim of a generalizable formal reasoning ability that is not studied in the paper.
> > > >
> > > > ## Additional suggestions
> > > >
> > > > It would be better if the authors could also provide the details about the calibration process between annotators as well as the compensation for them.
> > > >
> > > > I sincerely appreciate the authors' effort in responding to my suggestions. However, given the current evidence, I have decided to maintain my score.

---

> ### Author Response · Authors · 2025-12-03
> **Follow-up Response (Part I)**
>
> Thanks for your feedback. We address your concerns below.
>
> > Comment 1: Interleaving CoT is not a novel idea, neither training the model to do this interleaving.
>
> **Response 1:** We respectfully disagree with this assessment. While interleaving **reasoning** and **actions** has been explored in general-purpose LLM settings (e.g., ReAct), **there is no prior work in geometry problem solving that proposes or studies an interleaved** ***formal–natural*** **reasoning** **paradigm****.** Existing approaches fall into two disjoint categories:
>
> - **Pure formal reasoning** using symbolic programs only (e.g., GeoX, NGS, AlphaGeometry, GeoFormer), where the entire solution is expressed in a formal language expression.
> - **Pure natural-language reasoning** using multimodal LLMs, where the model generates only natural language CoT without executable formal expressions.
>
> As a result, existing methods either lack reasoning flexibility (formal-only) or lack mathematical precision (natural-language-only). Our work aims to bridge these two worlds.
>
> We greatly appreciate the conceptual inspiration from ReAct. However, we also emphasize that ReAct is not directly applicable here:
>
> - **Conceptually:** ReAct interleaves natural reasoning with natural language actions. Our method interleaves natural language reasoning with **formal symbolic operations**, which require strict syntax, mathematical rigor, and domain-specific semantics.
> - **Practically:** ReAct-style prompting does *not* work in our setting. Directly prompting SOTA models to mimic interleaved formal–NL reasoning yields **≤16.2% accuracy**, far below our **68.7%** post-trained model. Achieving successful interleaving requires **purpose-built data synthesis,** **SFT** **initialization, and solver-in-the-loop RL**, all of which prove essential in our ablations.
>
> > Comment 2: Why does it have to be a natural language and a specific formal language that are interleaved? Can it be any arbitrary formal language?
>
> **Response 2**: Thanks for sharing these questions.
>
> **Why do we interleave natural language and a specific formal language?**
>
> Natural language and formal language play complementary roles in geometry problem solving.
> - Natural language offers flexibility by auto-formalization and high-level planning. As shown in Figure 5, interleaving natural language with formal language provides better exploration potential, achieving much higher Pass@K scaling performance than pure formal reasoning.
> - Formal language enables precise symbolic manipulation and supports rigorous, step-level verification via external solvers. As shown in Table 3, compared with pure natural language, interleaved formal-natural CoT reduces computation errors to 0.3%, and significantly reduces reasoning errors by 14%.
>
> **Can it be any arbitrary formal language?**
>
> Not any arbitrary formal language would be equally effective. We believe the formal language should satisfy several properties:
> - Verifiability: It should be executable or checkable by an external solver.
> - Expressiveness: It should be capable of representing the key reasoning operations in the target domain.
> - Complementarity: It should provide capabilities that natural language lacks (e.g., precision, symbolic manipulation), rather than duplicating what natural language already does well.
> - Learnability: The syntax should be tractable for LLMs to acquire through fine-tuning
> Our choice of the formal language satisfies these criteria for geometric reasoning.
>
> We acknowledge that systematically studying which formal languages are most effective and whether these properties are sufficient conditions is an important direction for future work.

---

> ### Author Response · Authors · 2025-12-03
> **Follow-up Response (Part II)**
>
> > Comment 3: The claim that this training process helps models to "internalize formal reasoning patterns and seamlessly integrate them with natural language CoT" sounds like an overclaim.
> > 1. Another simplest explanation would be that we are teaching this new exotic language to the model, and perhaps this specific geometry solver language is suitable for the problems at hand, but may not be suitable for other problems.
> > 2. "Internalizing formal reasoning patterns" sounds like a broader claim of a generalizable formal reasoning ability that is not studied in the paper.
>
> **Response 3:**  Thank you for raising this concern. Our use of the term “internalize” follows the notion introduced in the work Learning by Distilling Context [1], where internalization refers to the process by which behaviors that initially require explicit contextual scaffolding (such as tools, examples, or templates provided in the prompt) become encoded in the model’s parameters through training, so that the model can subsequently deploy these behaviors without needing the original scaffolding.
>
> In our setting, training on interlevaed formal-natural language trajectories encourages the model to use the geometry solver language as an intermediate representation that it can autonomously produce and rely on when necessary when solving geometry problems.
>
> That said, we agree that our original phrasing in the former response could be read as making a broader claim about a general, domain-agnostic formal reasoning ability, which we do not empirically establish. To make this more rigorous and appropriately scoped, we rephrase our state:
>
> > "Our training procedure encourages the model to adopt the geometry solver language as an internal intermediate representation for the evaluated geometry tasks, which can then be combined with natural language CoT for reasoning"
>
> [1] Snell, Charlie, Dan Klein, and Ruiqi Zhong. "Learning by distilling context." arXiv preprint arXiv:2209.15189 (2022).
>
> > Comment 4: Additional suggestions: It would be better if the authors could also provide the details about the calibration process between annotators as well as the compensation for them.
>
> **Response 4:**
> 1. The calibration process between annotators. Prior to formal annotation, all annotators were taught the error taxonomy definitions. They discussed  ambiguous cases, and established consistent interpretation of the annotation. Then, annotators independently annotated a shared set of 300 examples. Throughout the annotation process, regular meetings were held to address emerging edge cases and maintain annotation consistency.
> 2. Compensation. All annotators were fairly compensated for roughly 0.5$ per problem, a standard rates for similar annotation tasks.

---

> ### Author Response · Authors · 2025-12-03
> **Response for Reviewer  VSGS (Part 1)**
>
> We sincerely thank for your response. We have addressed your questions below.
>
> > Comment 1: Interleaving CoT is not a novel idea, neither training the model to do this interleaving.
>
> **Response 1:** We respectfully disagree with this assessment. While interleaving **reasoning** and **actions** has been explored in general-purpose LLM settings (e.g., ReAct), **there is no prior work in geometry problem solving that proposes or studies an interleaved** ***formal–natural*** **reasoning** **paradigm****.** Existing approaches fall into two disjoint categories:
>
> - **Pure formal reasoning** using symbolic programs only (e.g., GeoX, NGS, AlphaGeometry, GeoFormer), where the entire solution is expressed in a formal DSL.
> - **Pure natural-language reasoning** using multimodal LLMs, where the model generates only natural language CoT without executable formal expressions.
>
> As a result, existing methods either lack reasoning flexibility (formal-only) or lack mathematical precision (natural-language-only). Our work aims to bridge these two worlds.
>
> We greatly appreciate the conceptual inspiration from ReAct. However, we also emphasize that ReAct is not directly applicable here:
>
> - **Conceptually:** ReAct interleaves NL reasoning with NL actions. Our method interleaves NL reasoning with **formal symbolic operations**, which require strict syntax, mathematical rigor, and domain-specific semantics.
> - **Practically:** ReAct-style prompting does *not* work in our setting. Directly prompting SOTA models to mimic interleaved formal–NL reasoning yields **≤16.2% accuracy**, far below our **68.7%** post-trained model. Achieving successful interleaving requires **purpose-built data synthesis, SFT initialization, and solver-in-the-loop RL**, all of which prove essential in our ablations.
>
> > Comment 2: Why does it have to be a natural language and a specific formal language that are interleaved? Can it be any arbitrary formal language?
>
> **Response 2:** Thanks for sharing these questions.
>
> **Why do we interleave natural language and a specific formal language?**
>
> Natural language and formal language play complementary roles in geometry problem solving.
>
> - Natural language offers flexibility by auto-formalization and high-level planning. As shown in Figure 5, interleaving natural language with formal language provides better exploration potential, achieving much higher Pass@K scaling performance than pure formal reasoning.
> - Formal language enables precise symbolic manipulation and supports rigorous, step-level verification via external solvers. As shown in Table 3, compared with pure natural language, interleaved formal-natural CoT reduces computation errors to 0.3%, and significantly reduces reasoning errors from 23.0% to 14.3%.
>
> **Can it be any arbitrary formal language?**
>
> Not any arbitrary formal language would be equally effective. We believe the formal language should satisfy several properties:
>
> - **Verifiability**: It should be executable or checkable by an external solver.
> - **Expressiveness**: It should be capable of representing the key reasoning operations in the target domain.
> - **Complementarity**: It should provide capabilities that natural language lacks (e.g., precision, symbolic manipulation), rather than duplicating what natural language already does well.
> - **Learnability**: The syntax should be tractable for LLMs to acquire through fine-tuning
>
> Our choice of the formal language satisfies these criteria for geometric reasoning. We acknowledge that systematically studying which formal languages are most effective and whether these properties are sufficient conditions is an important direction for future work.

---

> ### Author Response · Authors · 2025-12-03
> **Response for Reviewer VSGS (Part 2)**
>
> > Comment 3: The claim that this training process helps models to "internalize formal reasoning patterns and seamlessly integrate them with natural language CoT" sounds like an overclaim.
> >
> > 1. Another simplest explanation would be that we are teaching this new exotic language to the model, and perhaps this specific geometry solver language is suitable for the problems at hand, but may not be suitable for other problems.
> > 2. "Internalizing formal reasoning patterns" sounds like a broader claim of a generalizable formal reasoning ability that is not studied in the paper.
>
> **Response 3:**  Thank you for raising this concern. Our use of the term *“internalize”* follows the notion introduced in the work *Learning by Distilling Context [1]*, where *internalization* refers to the process by which behaviors that initially require explicit contextual scaffolding (such as tools, examples, or templates provided in the prompt) become encoded in the model’s parameters through training, so that the model can subsequently deploy these behaviors without needing the original scaffolding.
>
> In our setting, training on interlevaed formal-natural language trajectories encourages the model to use the geometry solver language as an intermediate representation that it can autonomously produce and rely on when necessary when solving geometry problems.
>
> That said, we agree that our original phrasing in the former response could be read as making a broader claim about a general, domain-agnostic formal reasoning ability, which we do not empirically establish. To make this more rigorous and appropriately scoped, we rephrase our state:
>
> > "Our training procedure encourages the model to adopt the geometry solver language as an internal intermediate representation for the evaluated geometry tasks, which can then be combined with natural language CoT for reasoning"
>
> [1] Snell, Charlie, Dan Klein, and Ruiqi Zhong. "Learning by distilling context." *arXiv* *preprint* *arXiv:2209.15189* (2022).
>
>
> >  Comment 4: Additional suggestions: It would be better if the authors could also provide the details about the calibration process between annotators as well as the compensation for them.
>
> **Response** **4****:**
>
> 1. The calibration process between annotators. Prior to formal annotation, all annotators were taught the error taxonomy definitions. They discussed  ambiguous cases, and established consistent interpretation of the annotation. Then, annotators independently annotated a shared set of 300 examples. Throughout the annotation process, regular meetings were held to address emerging edge cases and maintain annotation consistency.
> 2. Compensation. All annotators were fairly compensated for roughly 0.5$ per problem, a standard rates for similar annotation tasks.
>
> We hope these clarifications have addressed your concerns.

---

### Official Review · Reviewer_m5o1 · 2025-11-10

**Soundness:** 2
**Presentation:** 3
**Contribution:** 2
**Rating:** 6
**Confidence:** 3

**Summary:**

This work proposes a post-training strategy to teach Large Vision Language Models (LVLMs) to perform interleaved natural-language reasoning and formal reasoning, benifiting from the flexibility of natural language and the precision of formal language.
To do the post-training, authors synthesize an 11K-sample interleaved formal-natural language CoT dataset to conduct fine-tuning, and develop a solver-in-the-loop RL framework to further improve the performance.
Specifically, the interleaved formal-natural language CoT requires LVLMs first generate natural language reasoning and then generate symbolic programs step-by-step accordingly.
Authors claim that such interleaved and step-by-step program generation is better than direct generating the whole program.

**Strengths:**

1. Text, figures, tables are all good presented.
2. Experiments with various types of baselines and datasets well justify the performance of the proposed framework compared with existing methods.

**Weaknesses:**

1. Appendix B Related Work is not cited in the main text, which is an important part to understand the contribution.
2. The so-called "interleaved formal-natural language CoT" is suspicious. Figure 1 clearly shows that the interleaved CoT involves interleaved natural-language reasoning statement and formal reasoning statement. However, in Appendix D, the so-called interleave CoT is more like a three-stage reasoning process, i.e., step-by-step natural-language reasoning, generating formal statements step-by-step according to the natural-language reasoning steps, and aggregating the formal statements to form a program. This raises the concerns that
    - The definition of "interleaved formal-natural language CoT" is not consistent across the whole paper.
    - The key innovation is adding a step-by-step formal statement generation between vanilla CoT and program generation, which seems not as interesting as Introduction claims.
3. Since the motivation of generating interleaved formal-natural language CoT is that directly generating the program is prone to errors, a comparison between the current framework with the one without interleaved formal-natural language CoT (keep fine-tuning, RL, and any other component in the current framework) is needed.

**Questions:**

1. How about the performance without fine-tuning? Can the proposed solver-in-the-loop RL framework work without fine-tuning?
2. Why the proposed reasoning process is called interleaved formal-natural language reasoning but the proposed dataset is called formal-interleaved CoT dataset (without *natural*) ?


Typo:
1. Line 228-229: $\hat{r}$ should be $\hat{z}$
2. Line 236-237: $z$ should be $x$?

---

> ### Author Response · Authors · 2025-11-24
> **Response**
>
> Thank you for taking the time to read our paper and for providing a positive review of our work. We appreciate your thoughtful feedback. Below, we address your concerns and answer your questions.
>
> **Comment 1:** Appendix B Related Work is not cited in the main text, which is an important part to understand the contribution.
> **Response 1:** Thank you for pointing this out. We agree that the Related Work section is important for understanding our contributions. With the one-page extension allowed during the rebuttal period, we have moved the Related Work from Appendix B into the main text in the revised manuscript.
>
>
> **Comment 2:** The so-called "interleaved formal-natural language CoT" is suspicious. Figure 1 clearly shows that the interleaved CoT involves interleaved natural-language reasoning statement and formal reasoning statement. However, in Appendix D, the so-called interleave CoT is more like a three-stage reasoning process, i.e., step-by-step natural-language reasoning, generating formal statements step-by-step according to the natural-language reasoning steps, and aggregating the formal statements to form a program.
>
>
> **Response2:** We thank the reviewer for their careful examination of our appendices. We would like to clarify the distinction between our demonstration prompts and the actual generated outputs.
>
> The "three-stage reasoning process" you observed may refer to the **manually-written demonstration prompts** in Appendices D.1-D.2, not the model-generated CoT. We designed these prompts with an initial planning phase (natural language problem analysis and planning) before the interleaved reasoning begins. This structured approach helps guide MLLMs to generate more successful interleaved trajectories during inference.
>
> The synthetic data and **model outputs** consistently produce genuinely interleaved CoT consistent with Figure 1, where natural language and formal geometric operators alternate throughout the reasoning chain. This can be verified in:
>
> - Our synthetic CoT dataset examples in Appendix D.4.
> - The model-generated solutions shown in the green boxes of Appendix C's Computation Error Case and Reasoning Error Case.
>
> **Comment 3:** Since the motivation of generating interleaved formal-natural language CoT is that directly generating the program is prone to errors, a comparison between the current framework with the one without interleaved formal-natural language CoT (keep fine-tuning, RL, and any other component in the current framework) is needed.
>
> **Response 3:** Thanks for this suggestion. We agree that demonstrating the necessity of formal-natural interleaved CoT is necessary. We would like to clarify that our paper already includes relevant comparisons, and we have conducted additional experiments as you suggested:
>
> - **Relevant Comparisons in the Submission:**
>   - **Interleaved CoT vs. Pure formal Solution****:** In the first paragraph of Section 4.3 **(Section 3.3 in the submission version)**, we have compared SFT models trained on interleaved CoT versus pure formal solutions (directly generating programs without natural language). As shown in Figure 5, pure formal reasoning demonstrates inferior Pass@K scaling, with the performance gap widening as the number of samples increases. For instance, on the UniGEO dataset, Interleaved CoT yields a 3.2% improvement in Pass@1 and a more substantial 13.7% improvement in Pass@8. **This illustrates that interleaved CoT provides better exploration potential.**
>   - **Interleaved CoT vs. Natural language CoT:** In the second paragraph of Section 3.3, we compared SFT+RL models using interleaved CoT versus pure natural language CoT. As shown in Table 3, interleaved formal-natural CoT demonstrates **reductions in reasoning errors, geometry knowledge errors, and computation errors** compared to natural language alone.
>
> - **Additional Comparisions (Following Your Suggestion):**
>   - To provide a more direct comparison, we conducted comparisons where all three reasoning paradigms use the same training dataset and identical SFT + RL training procedures. These results demonstrate that interleaved CoT consistently outperforms both pure formal and pure natural language approaches, validating that the interleaving mechanism is essential rather than merely adding intermediate steps.
>
> | **Reasoning** **Paradigm**                    | PGPS9K    | UniGEO    |
> | --------------------------------------------- | --------- | --------- |
> | Pure Formal Language Reasonong                | 51.8%     | 55.2%     |
> | Pure Natural  Language Reasoning              | 60.0%     | 68.7%     |
> | **Interleaved Formal-Natural CoT** **(Ours)** | **68.7%** | **72.7%** |
>
> We have added the comparisons in Appendix C's Reasoning Paradigm Comparison for clarification.

---

> > ### Author Response · Authors · 2025-11-24
> > **Response for Questions**
> >
> > **Question 1:** How about the performance without fine-tuning? Can the proposed solver-in-the-loop RL framework work without fine-tuning?
> >
> > **Response 2**: Thank you for this question. The short answer is **no**—the solver-in-the-loop RL framework cannot function without (supervised) fine-tuning for addressing cold-start. The base model, lacking exposure to formal language syntax, would default to using natural language for problem-solving and fail to generate the executable formal expressions required by the solver. As a result, the solver-in-the-loop reinforcement learning process cannot be properly initiated, and the accuracy remains at zero throughout.
> >
> >
> > **Question 2**: Why the proposed reasoning process is called interleaved formal-natural language reasoning but the proposed dataset is called formal-interleaved CoT dataset (without *natural*) ?
> >
> > **Response 2**:  Thank you for pointing out this terminology inconsistency. You are correct that the naming could be clearer. The discrepancy arose because:
> >
> > - In the literature, "CoT" (Chain-of-Thought) conventionally refers to natural language reasoning by default. Therefore, when we initially used "formal-interleaved CoT",  we implicitly meant "formal statements interleaved with natural language CoT".
> > - However, we recognize this implicit assumption creates ambiguity and may confuse readers.
> >
> > To address this issue, we have **standardized the terminology** throughout the revised manuscript to consistently use **Interleaved Formal-Natural  CoT** (or "Interleaved  CoT") for both the reasoning process and the dataset. This ensures clarity and eliminates any potential confusion about whether natural language is included in the interleaving.
> >
> >
> > **Response for Typos:** Thank you for catching these typos. We have corrected them in the main text.
> >
> > We sincerely thank you for your thoughtful and constructive feedback. We hope the clarifications and additional analyses in our revised manuscript have addressed your concerns. If so, we would be grateful if you could consider updating your review score accordingly. Please do not hesitate to let us know if you have any further questions.

---

### Author Response · Authors · 2025-11-24
**General Response and Summary of Changes**

We sincerely thank all reviewers and area chairs for their efforts in reviewing our paper and providing valuable feedback. We have carefully addressed each concern raised by the reviewers and made corresponding revisions to our paper. We highlight the revision in blue. Below, we summarize the key changes in our revision:
- **Section 2 (Page 3)**: We have moved the related work section from the Appendix to the main text  to provide better context for understanding our contributions.
- **Appendix B**: We added the discussion with the distinction with ReACT and Thinking with Images.
  - **Comparison with ReACT**. We added the discussion with the distinction with ReACT. Our method has fundamental differences in problem setting and technique. ReACT is an inference-time prompting framework that assumes pre-existing reasoning capabilities, while ours uses post-training to instill interleaved reasoning and formal operator skills absent in base models. Moreover, ReACT interleaves natural language reasoning with natural language actions, whereas our method combines natural language with formal symbolic operations, demanding stricter syntactic correctness.
  - **Comparison with Thinking with Images**. Our approach is orthogonal and complementary to "Thinking the Images". While ”Thinking with Images” emphasizes visual representations and planning in the image space, leveraging visual reasoning for geometric understanding. Our approach proposes a new reasoning paradigm in the language space that interleaves formal symbolic computation with natural language reasoning, aiming to achieve both mathematical precision (through formal language) and reasoning flexibility (through natural language).
- **Appendix C**: We have provided addtional experimental results and explanations below.
  - **Reasoning Paradigm Comparison (Page 15, Table 6)**. Table 6 compares three reasoning approaches: pure formal language, pure natural language, and interleaved formal-natural CoT. The results demonstrate that interleaved CoT consistently outperforms both pure approaches, validating the benefits of interleaving for improving performance.
  - **Performance Gains Beyond Tool Access (Page 15, Table 7)**. Table 7 isolates the performance gains attributed to tool access. The results show that even when given access to tools, strong foundation models still struggle to generate valid interleaved CoT that the solver can execute, resulting in low accuracy. This demonstrates that our performance gains cannot be attributed to tool access alone, but rather stem from teaching models to internalize solver knowledge and reasoning patterns.
  - **SFT Epoch Influence on More Benchmarks (Page 16, Figure 8)**. Figure 8 shows that while SFT is important for building fundamental capacity, excessive SFT may compromise response diversity, consequently limiting the model's ability to establish reasoning capacity during RL training. This phenomenon is consistently observed across multiple benchmarks, including UniGEO, MathVista, and MathVerse.
  - **On the Design of the Reward Function (Page 17, Figure 10)**. Figure 10 presents the results of adding a process reward during RL training. The results show that process rewards do not yield additional gains compared to using the binary outcome reward alone, which may be due to the fact that through multiple rollouts in the online RL stage, outcome reward can provide implicit partial credit.
  - **Comparison with Recent Frontier Models (Page 17, Table 9)**.  Table 9 compares our approach with recent frontier models, including Qwen3-VL-8B-Instruct, GLM-4.5V, Gemini 2.5 Pro, GPT-5, and the "Thinking with Images" model (e.g., OpenAI o3). The results show that our 32B-version model achieves competitive performance with significantly larger commercial models. Furthermore, scaling from 7B to 32B yields approximately 14% performance improvement, suggesting that further gains are possible with larger base models.
  - **Zooming Images for Better Perception (Page 18, Figure 11)**. Figure 11 provides a supporting case where applying a 2x zoom enables the model to obtain a better visual perception. The result indicates image-processing tools aid visual reasoning and complement the current framework.
  - **Error Analysis: Comparing Interleaved Formal-Natural vs. Natural Language Reasoning (Page 19)**. We have added detailed illustrations of the four types of errors identified in our analysis, along with annotation details and agreement statistics.

We believe the revisions outlined above have significantly enhanced the quality of our paper, thanks to the insightful feedback from the reviewers. We hope these updated results adequately address the reviewers' concerns and further strengthen the contributions of our work. Thank you once again for your valuable feedback.

---

### Author Response · Authors · 2025-12-03
**# Summary of Response**

## Summary of Reviewer Concerns and Author Responses

We thank the Area Chairs and all reviewers for their time and constructive feedback! We summarize below the main concerns and how the revised manuscript addresses them.

> **Concern 1: Necessity and novelty of the interleaved formal–natural CoT** **paradigm** *(raised by Reviewers  m5o1* *and VSGS**)*

**Response:** In the revised manuscript, we clarify this by:

- **Adding controlled comparisons** showing that interleaved CoT consistently outperforms both pure natural-language CoT and pure formal-language programs under the same training conditions (+8.7% on PGPS9K and +4.0% on UniGEO).
- Explaining how our setting differs fundamentally from ReAct and tool-use frameworks: our work requires **learning new formal symbolic capabilities**, not orchestrating existing skills.
- Standardizing terminology and revising related-work discussions to clearly position our paradigm and contributions within the literature.

These results collectively illustrate that interleaving is **both necessary** (because neither pure modality suffices) and **novel** in the context of domain-specific geometry problem solving.

> **Concern 2: Fairness of comparisons and potential confounding factors (e.g., operator set, tool access, teacher model)** *(raised by Reviewers VSGS, 8pkS, and 2ABk)*

**Response:**  We expanded our analyses to explicitly disentangle model improvements from confounding factors:

- **Tool access alone does not explain performance**. Strong models (GPT-4o, Claude-Sonnet, Qwen-VL) given the *exact same* operator specifications still fail to generate valid interleaved CoT.
- **Teacher-model dependency issue**: teacher trajectories are filtered through a solver; RL training is fully teacher-independent; and our model **significantly surpasses** the teacher model itself (68.7% vs. 37.6%) in interleaved thinking.
- **Operator-set** **scaling** **is orthogonal**: our framework is *designed to be scalable*, and the current 34-operator set matches standard practice in prior geometry solvers. We clarify this in the text.

These changes ensure that reported gains originate from the **interleaved reasoning** **paradigm** **and post-training pipeline**, rather than from external resources or privileged access.

> **Concern 3: Completeness of technical analysis (training dynamics, SFT/RL behavior, ablations, dataset clarity)** *(raised by Reviewers m5o1, VSGS, and 2ABk)*

**Response:**  We expanded our technical evaluation and added new ablations:

- **SFT epoch analysis** across UniGEO, MathVista, and MathVerse shows overfitting trends and explains why longer SFT weakens RL performance due to entropy collapse.
- **Reward-design ablations** compare binary rewards with process-level partial-credit rewards, showing no improvement (and slight degradation), consistent with findings from prior work.
- **Additional controlled comparisons** across three reasoning paradigms (pure formal, pure NL, interleaved) under identical training setups.
- **Additional comparison with recent frontier models** shows our 32B model achieves competitive performance with state-of-the-art proprietary models (Gemini 2.5 Pro, GPT-5), despite being significantly smaller.
- Expanded explanations and examples in Appendix C to fully illustrate error categories and analyses.

These revisions directly address all remaining technical and clarity concerns.

## Summary for AC

Overall, we believe the remaining concerns are primarily about **clarification and presentation**, rather than fundamental flaws in the approach. Reviewer reactions after rebuttal are generally positive:

- **Reviewer m5o1** assigned a score of **6** in the initial review and expressed clear support for the work.
- **Reviewer 2ABk** acknowledged that our rebuttal addressed their concerns and **raised their score from 4 to 6**, explicitly signaling positive support.
- **Reviewer VSGS** provided further detailed feedback after the rebuttal, which we have addressed point-by-point in this revision. We believe our responses have adequately clarified the concerns raised.
- **Reviewer 8pkS** confirmed having read the rebuttal, and although no additional comments were provided, he or she rated the paper as *“good”* in soundness and presentation, and offered positive recognition of our work—noting the "excellent performance and efficiency," the effectiveness of our "SFT + solver-in-the-loop RL" framework, and the "valuable insights" provided by our ablation studies. We interpret this as supportive.

---

> ### Author Response · Authors · 2025-12-03
> **Summary of Response (Continued)**
>
> ## Final Contributions
>
> The revised paper presents:
> - **A new paradigm**: Interleaved formal–natural CoT, showing superior performance, exploration potential, and reasoning preciseness compared to prior approaches.
> - **A new cold-start dataset**: The first interleaved formal–natural CoT dataset for geometry reasoning,  with careful designs from data patterns and model responses, filling an important gap in the community.
> - **A comprehensive study of post-training strategies**: Detailed analysis of how SFT and solver-in-the-loop RL jointly enable robust symbolic-natural reasoning.
>
> We hope the above clarifications and improvements assist the Area Chair in making a fair final decision. Thank you again for your time and consideration.

---

### Note · Program_Chairs · 2026-01-17
**Submission Desk Rejected by Program Chairs**

The following references in this submission do not refer to real documents and/or have major errors in bibliographic information:

     Jiaqi Chen, Jiazhan Yan, Xin Lin, Jian Dong, Ziqi Yuan, and Ji-Rong Lu. Geoqa: A geometric question answering benchmark towards multimodal mathematical reasoning. arXiv preprint arXiv:2303.03242, 2023.